# *Integrin α7* is a functional cancer stem cell surface marker in oesophageal squamous cell carcinoma

Xiao-Yan Ming[1,2], Li Fu[3], Li-Yi Zhang[1,2], Yan-Ru Qin[4], Ting-Ting Cao[1,3], Kwok Wah Chan[2,5], Stephanie Ma[2,6], Dan Xie[7] & Xin-Yuan Guan[1,2,7]

Non-CG methylation has been associated with stemness regulation in embryonic stem cells. By comparing differentially expressed genes affected by non-CG methylation between tumour and corresponding non-tumour tissues in oesophageal squamous cell carcinoma (OSCC), we find that *Integrin α7* (*ITGA7*) is characterized as a potential cancer stem cell (CSC) marker. Clinical data show that a high frequency of ITGA7$^+$ cells in OSCC tissues is significantly associated with poor differentiation, lymph node metastasis and worse prognosis. Functional studies demonstrate that both sorted ITGA7$^+$ cells and ITGA7 overexpressing cells display enhanced stemness features, including elevated expression of stemness-associated genes and epithelial–mesenchymal transition features, as well as increased abilities to self-renew, differentiate and resist chemotherapy. Mechanistic studies find that ITGA7 regulates CSC properties through the activation of the FAK-mediated signalling pathways. As knockdown of *ITGA7* can effectively reduce the stemness of OSCC cells, ITGA7 could be a potential therapeutic target in OSCC treatment.

[1] Department of Clinical Oncology, Li Ka Shing Faculty of Medicine, The University of Hong Kong, Hong Kong 852, China. [2] Centre for Cancer Research, Li Ka Shing Faculty of Medicine, The University of Hong Kong, Hong Kong 852, China. [3] Department of Pharmacology, Shenzhen Key Laboratory of Translational Medicine of Tumor and Cancer Research Centre, School of Medicine, Shenzhen University, Shenzhen 518000, China. [4] Department of Clinical Oncology, The First Affiliated Hospital, Zhengzhou University, Zhengzhou 450052, China. [5] Department of Pathology, Li Ka Shing Faculty of Medicine, The University of Hong Kong, Hong Kong 852, China. [6] School of Biomedical Sciences, Li Ka Shing Faculty of Medicine, The University of Hong Kong, Hong Kong 852, China. [7] State Key Laboratory of Oncology in Southern China, Sun Yat-Sen University Cancer Center, Guangzhou 510060, China. Correspondence and requests for materials should be addressed to X.-Y.G. (email: xyguan@hku.hk).

Oesophageal squamous cell carcinoma (OSCC), the major histologic subtype of oesophageal cancer, is one of the most common malignancies and ranked as the sixth leading cause of cancer death worldwide[1]. OSCC is characterized by its remarkable geographic distribution with particularly high-risks in Northern China[2]. Despite advances in diagnosis and treatment of OSCC, the 5-year survival rate after curative surgery is only 20–30%, mainly due to tumour metastasis, tumour recurrence and chemoresistance[1]. Recently, increasing evidence suggests that cancer stem cells (CSCs) represent an important subset of tumour cells that are responsible for tumour metastasis, tumour recurrence and chemoresistance[3–6]. CSCs are a subset of cancer cells that are biologically distinct from the others and have stem cell-like features[7]. CSC has now been showed to exist in different tumour types including breast cancer[8], hepatocellular carcinoma[9] and OSCC[10]. Currently, most CSCs are marked by either cell surface markers (for example, CD90 or CD133) or functional properties (for example, site population)[11]. In OSCC, surface proteins CD90 (ref. 10), CD44 (ref. 12), and p75[NTR] (ref. 13), have previously been used to phenotypically mark and functionally define oesophageal CSC subsets in cell lines and clinical samples. These markers can markedly enrich the oesophageal CSC subpopulation, however, the purity of CSCs is unknown. In addition, the roles of these markers in stemness regulation and CSC maintenance are rarely understood.

A recent study by Lister et al. found that nearly 25% of all methylated sites identified in embryonic stem cells (ESC) are in a non-CG context[14]. Non-CG methylation disappears on induced differentiation of ESCs and can be restored in induced pluripotent stem cells (iPSC)[14]. Another report also demonstrated that non-CG methylation appears to be confined to stem cells[15]. As CSCs share many common properties with stem cells[16,17], we hypothesize that non-CG methylation also predominantly exists in CSCs but not in other differentiated cancer cells. To test this, we first selected 10 stemness-associated genes with differential non-CG methylation between ESC and fibroblast from Lister et al.'s report for preliminary study in OSCC cell lines and clinical specimens. Among them, integrin α7 (ITGA7) was chosen for further study because it was only expressed in a small group of cancer cells in both clinical specimens and cell lines. Integrins are transmembrane cell surface receptors comprised of 18 α and 8 β subunits. Integrins directly bind components of the extracellular matrix (ECM) and provide the traction necessary for cell motility and invasion. ITGA7, the receptor for the ECM protein laminin, forms heterodimer with integrin β1. Integrins are involved in a broad spectrum of cellular processes, including survival, proliferation, migration and invasion[18]. Recent studies have suggested that integrins play important roles in the regulation of stem cell-like properties and enrich CSCs from glioblastoma multiforme (GBM)[19], breast cancer[20] and prostate cancer[21].

In the present study, we find that ITGA7[+] cells to be sporadically expressed in clinical OSCC specimens and that the presence of ITGA7[+] cells is closely correlated with aggressive tumour behaviour and poor outcome. Functional studies demonstrate that ITGA7[+] cells possess strong CSC properties including abilities to self-renew, differentiate and resist chemotherapy. The molecular mechanism by which ITGA7 regulates these properties is also elucidated. Our data indicate that ITGA7 is a potential CSC marker in OSCC with function in stemness regulation.

## Results

**ITGA7[+] cells are markedly associated with poor outcome.** Expression of ITGA7 was analyzed by immunohistochemistry (IHC) on a tissue microarray (TMA) consisting of 300 paired OSCC and non-tumour clinical samples. Informative IHC results

were obtained from 262 pairs of OSCCs. Non-informative samples included lost samples and unrepresentative samples, which were not included in data complication. ITGA7-expressing cells were detected in most of these informative OSCC cases, with expression ranging from 0 to 5%; while ITGA7 expression could not be detected in any of the corresponding non-tumour tissues (Fig. 1a). On the basis of the frequency of ITGA7 positive cells (ITGA7[+]), the OSCC patients were almost equally divided into high-frequency group ($>0.6\%$, $n = 137$, 52.3%) and low-frequency group ($\leq 0.6\%$, $n = 125$, 47.7%) (Supplementary Table 1). Association study found that the high-frequency group was significantly associated with poor differentiation (Pearson $\chi^2$ test, $P = 0.001$), presence of invasion (Pearson $\chi^2$ test, $P = 0.009$), advanced clinical stage (Pearson $\chi^2$ test, $P < 0.001$) and lymph node metastasis (Pearson $\chi^2$ test, $P = 0.005$; Table 1). Kaplan–Meier survival analysis based on this TMA data found that OSCC patients with high frequency of ITGA7[+] cells ($>0.6\%$) were significantly associated with a shorter survival time (log-rank test, $P < 0.001$; Fig. 1b). Next, we used flow cytometry to detect the frequency of ITGA7[+] cells in 2 immortalized oesophageal epithelial cell lines (NE1 and NE3) and 10 OSCC cell lines. The result showed that the frequency of ITGA7[+] cells in OSCC cell lines (except EC109) was much higher than that in immortalized oesophageal epithelial cell lines (Fig. 1c; Supplementary Fig. 1a). This result was further confirmed by immunofluorescence in NE1, EC109, KYSE180 and KYSE520 cell lines (Supplementary Fig. 1b).

**Non-CG methylation is mainly detected in ITGA7[+] OSCC cells.** Since non-CG methylation plays important roles in the transcriptional regulation of target genes in stem cells, we next investigated whether ITGA7 expression was up-regulated in this manner. Three DNA fragments within ITGA7 gene containing CG and non-CG methylation loci were selected based on epigenetic modification data between ESC and fetal fibroblasts (http://neomorph.salk.edu/human_methylome). Bisulfite genomic sequencing (BGS) analysis was performed in sorted ITGA7[+] and ITGA7[−] cells from KYSE180 and KYSE520 cell lines. The result showed that ITGA7[+] cells displayed much higher non-CG methylation frequency than their negative counterparts, implying that ITGA7 expression might be correlated with non-CG methylation (Fig. 1d). To determine whether the expression of ITGA7 could be restored by DNA demethylation, a DNA methylation inhibitor 5-aza-2′-deoxycytidine was used to treat sorted ITGA7[+] and ITGA7[−] cells. After treatment, expression of ITGA7 was significantly increased (Fig. 1e).

**ITGA7 and CD90 is co-expressed in OSCC.** CD90 has previously been reported by our group to phenotypically mark a functionally defined oesophageal CSC subset[10]. To examine whether CD90 and ITGA7 is co-expressed, dual-colour flow cytometry was performed to examine their expression in a panel of oesophageal cell lines (Supplementary Fig. 2). Our findings included: (1) the frequencies of ITGA7[+] cells and CD90[+] cells were in the same trend (Fig. 2a; Supplementary Table 2); (2) ITGA7[+] cells were much less abundant than CD90[+] cells in all oesophageal cell lines; and (3) most ITGA7[+] cells in OSCC cell lines were also CD90[+] (Supplementary Table 3). To confirm that ITGA7 was co-expressed with CD90, cryosectioned spheroids generated from KYSE520 cells were stained and analyzed by immunofluorescence. The result showed that most of ITGA7[+] cells were co-expressed with CD90 (Fig. 2b). These findings suggested that ITGA7[+] cells may represent and mark a more specific oesophageal CSC subset than CD90. To validate this, CD90[+]/ITGA7[+], CD90[+]/ITGA7[−] and CD90[−]/ITGA7[−] cells were separated from KYSE520 cells by cell sorting, while

CD90$^-$/ITGA7$^+$ subpopulation does not exist because most ITGA7$^+$ cells are also CD90$^+$ (Fig. 2c). Western blot analysis demonstrated that the expression of stemness-associated genes was markedly higher in CD90$^+$/ITGA7$^+$ cells than that in CD90$^+$/ITGA7$^-$ or CD90$^-$/ITGA7$^-$ cells (Fig. 2d). Spheroid formation assay also found that the self-renewal ability was significantly enhanced in CD90$^+$/ITGA7$^+$ cells, compared with CD90$^+$/ITGA7$^-$ and CD90$^-$/ITGA7$^-$ cells (Fig. 2e). In addition, CD90$^+$/ITGA7$^+$ cells were able to induce more and bigger colonies than those induced by CD90$^+$/ITGA7$^-$ and CD90$^-$/ITGA7$^-$ cells (Fig. 2f).

**ITGA7$^+$ OSCC cells possess CSC properties.** To further confirm ITGA7$^+$ OSCC cells are CSCs, major cancer and stemness properties including expression of stemness-associated genes and abilities to self-renew, differentiate and resist chemotherapy were analyzed in ITGA7 sorted subsets isolated from KYSE180 and KYSE520 cells. ITGA7$^+$ cells expressed higher levels of OCT3/4, SOX2, NANOG and CD90, compared with ITGA7$^-$ cells (Fig. 3a). Further, ITGA7$^+$ cells displayed significantly enhanced ability to initiate spheroid formation and to expand in subsequent serial propagations, compared with ITGA7$^-$ cells, demonstrating their increased ability to self-renew (Fig. 3b).

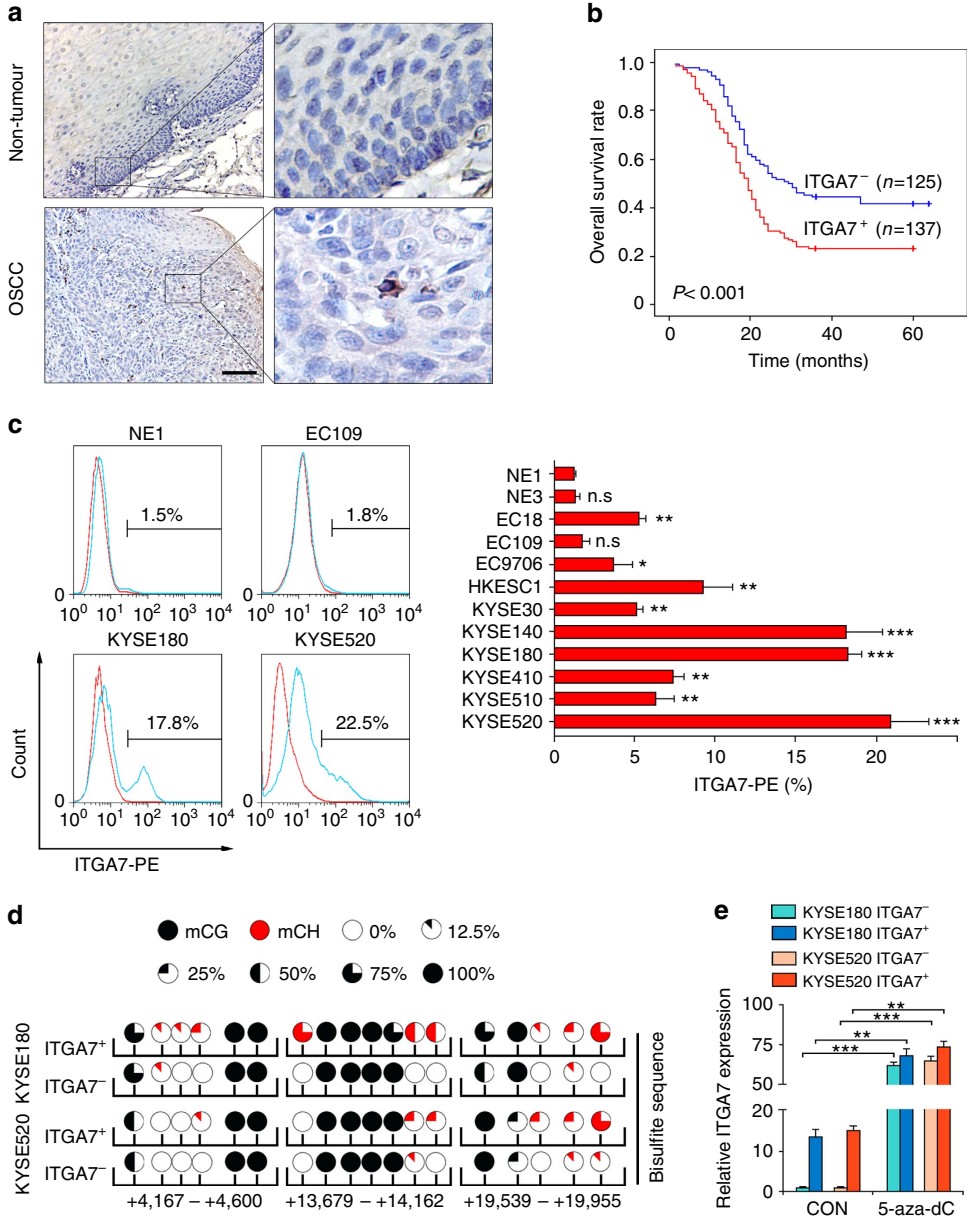

**Figure 1 | High frequency of ITGA7$^+$ cells is significantly associated with poor outcome in OSCC.** (**a**) Representative IHC images show that ITGA7$^+$ cells were scattered in OSCC tumour tissue in clinical specimen, but not in non-tumour tissue. Scale bar, 100 μm. (**b**) Kaplan–Meier survival analysis shows that OSCCs with high frequency of ITGA7$^+$ cells ($>0.6\%$, ITGA7$^+$, $n=137$) had shorter survival time, compared with OSCCs with low frequency of ITGA7$^+$ cells ($\leq 0.6\%$, ITGA7$^-$, $n=125$). (**c**) Percentage of ITGA7$^+$ cells detected by FACS in immortalized esophageal epithelial and OSCC cell lines. The average percentage of ITGA7$^+$ cells, the mean ± s.d. of three independent detections, in different cell lines was depicted in the bar chart. (**d**) Detection of DNA methylation in the CG and CH context (H = A, C or T) by genomic bisulfite sequence. Non-CG methylation of ITGA7 preferred to occur in ITGA7$^+$ cells isolated from KYSE180 and KYSE520. (**e**) qRT-PCR showed that the expression of ITGA7 was markedly increased after treated with 5-aza-2′-deoxycytidine (5-aza-dC, 50 μM) for 3 days. Statistics: (**c,e**) ANOVA with *post hoc* test. *$P<0.05$; **$P<0.001$; ***$P<0.0001$; n.s., $P\geq 0.05$.

To investigate the differentiation potential of ITGA7$^+$ cells, differentiation inducer all-trans retinoic acid (atRA) was used to treat ITGA7$^+$ and ITGA7$^-$ cells. Western blot analysis showed that atRA treatment could effectively induce the differentiation of ITGA7$^+$ cells by increasing CK13 (a marker of differentiated supra basal cells) and decreasing CK14 (a marker of oesophageal basal cell) expression. Consequently, the expression of ITGA7 and stemness-associated genes OCT3/4, SOX2 and NANOG was also decreased (Fig. 3c). However, no expression change was observed in ITGA7$^-$ cells after atRA treatment. This data demonstrated that ITGA7$^+$ cells possessed differentiation ability. To check whether ITGA7$^+$ cells are more chemoresistant, chemotherapeutic reagents cisplatin and 5-fluorouracil (5-FU) were used to treat ITGA7$^-$ and ITGA7$^+$ OSCC cells. Flow cytometry analysis showed that the apoptotic/necrotic cells were significantly higher in ITGA7$^-$ cells compared with ITGA7$^+$ cells (Fig. 3d), indicating that ITGA7$^+$ cells were more resistant to the chemotherapeutic drugs. To further investigate the *in vivo* tumorigenic potential of ITGA7$^+$ cells, ITGA7$^+$ and ITGA7$^-$ cells were subcutaneously engrafted in NOD/SCID mice, respectively. As few as 10,000 ITGA7$^+$ cells were sufficient to generate visible tumours 4 months post injection, whereas, at least 100,000 ITGA7$^-$ cells were necessary to generate a tumour in the same mouse model (Fig. 3e; Table 2).

**Knockdown of *ITGA7* reduces CSC properties in OSCC cells.** To further investigate if ITGA7 is a functional marker of oesophageal CSCs, we performed similar assays to examine cancer and stemness properties in OSCC cells with ITGA7 stably suppressed. Expression of ITGA7 in KYSE180 and KYSE520 cells was silenced with *ITGA7*-specific short hairpin RNA (shRNA). Cells treated with scrambled shRNA were used as non-target controls (NTC). After confirmation of successful knockdown with a 60% reduction by flow cytometry (Fig. 3f) and western blotting (Fig. 3g), we first detected the expression of stemness-associated genes. Compared with cells treated with scrambled shRNA, OCT3/4, SOX2 and NANOG were down-regulated in sh*ITGA7*-treated cells (Fig. 3g). To determine whether *ITGA7* knockdown could affect chemoresistant potential, cisplatin and 5-FU were used to treat sh*ITGA7*-transfected cells. The apoptotic index of sh*ITGA7*-transfected OSCC cells was significantly higher than that of control cells after a 48-h exposure to chemotherapeutic reagents (Fig. 3h; Supplementary Fig. 3). Spheroid formation assay showed that sh*ITGA7*-transfected KYSE180 and KYSE520 cells formed smaller and less spheroids, compared with control cells (Fig. 3i). In view of the importance of ITGA7 in regulating differentiation, western blot analysis was used to detect expression level of well differentiation marker CK13 and poor differentiation marker CK14 in OSCC. The result showed that ITGA7 could up-regulate CK14 and down-regulate CK13 expression (Fig. 3j). To investigate the *in vivo* tumorigenic potential of ITGA7 knockdown cells, scrambled shRNA and sh*ITGA7*-treated cells were subcutaneously engrafted in NOD/SCID mice, respectively. To minimize the number of mice to be used, two *ITGA7*-specific shRNAs-treated cells were equally mixed together as sh*ITGA7* group. Compared with control group, knockdown of *ITGA7* could effectively suppress tumour initiation (Table 3).

**Ectopic overexpression of ITGA7 enhances CSC properties.** As a complementary model, *ITGA7* was also stably transfected into EC109 and KYSE30 cells. Expression level of ITGA7 in *ITGA7*-transfected cells was determined by western blotting (Fig. 4a) and flow cytometry (Fig. 4b). Western blotting results also showed that the ectopic expression of ITGA7 could enhance

**Table 1 | Association of ITGA7 expression with clinicopathologic features in 262 primary OSCC cases.**

| Features | Total | ITGA7 expression* | | P-value[†] |
|---|---|---|---|---|
| | | Low frequency | High frequency | |
| *Sex* | | | | 0.390 |
| Male | 150 | 75 (50%) | 75 (50%) | |
| Female | 112 | 50 (44.6%) | 62 (55.4%) | |
| *Age* | | | | 0.593 |
| <60 | 138 | 68 (49.3%) | 70 (50.7%) | |
| ≥60 | 124 | 57 (46%) | 67 (54%) | |
| *Differentiation* | | | | 0.001[‡] |
| Well/moderate | 195 | 105 (53.8%) | 90 (46.2%) | |
| Poor | 67 | 20 (29.9%) | 47 (70.1%) | |
| *Tumour stage[§]* | | | | <0.001[‡] |
| Early stage I-II | 170 | 96 (56.5%) | 74 (43.5%) | |
| Advanced stage III-IV | 92 | 29 (31.5%) | 63 (68.5%) | |
| *Invasion* | | | | 0.009[‡] |
| Absent | 90 | 53 (58.9%) | 37 (41.1%) | |
| Present | 172 | 72 (41.9%) | 100 (58.1%) | |
| *Lymph-node metastasis* | | | | 0.005[‡] |
| Absent | 142 | 79 (55.6%) | 63 (44.4%) | |
| Present | 120 | 46 (38.3%) | 74 (61.7%) | |

*ITGA7 percentage ≤0.6% as low frequency; >0.6% as high frequency.
[†]Pearson $\chi^2$ test.
[‡]Significance.
[§]AJCC/UICC TNM staging system.

expression of stemness-associated genes (Fig. 4a). To examine whether the ectopic expression of ITGA7 could affect chemoresistant potential, cisplatin and 5-FU were used to treat *ITGA7*-transfected cells. The apoptotic index of *ITGA7*-transfected cells was significantly lower than that of control cells, which is similar to sorted ITGA7$^+$ cells (Fig. 4c). Spheroid formation assay showed that *ITGA7*-transfected cells formed significantly larger and more spheroids, compared with control cells (Fig. 4d). Ectopic overexpression of ITGA7 resulted in a marked decrease in the well-differentiated marker CK13. Note EC109 did not express CK13 and neither cell lines expressed CK14, and thus deregulation in expression could not be tested (Fig. 4e).

The tumorigenic ability of ITGA7 was assessed by foci formation and soft agar assays. Compared with control cells, *ITGA7*-expressing cells displayed higher foci formation frequencies (Fig. 4f) and enhanced colony forming abilities in soft agar (Fig. 4g). To further investigate the *in vivo* tumorigenic ability of ITGA7, *ITGA7*-transfected cells and control cells were subcutaneously injected into the right and left dorsal flank of NOD/SCID mice, respectively. The result found that *ITGA7*-overexpressed EC109 and KYSE30 cells displayed significantly stronger tumour initiation ability than their control cells (Fig. 4h; Table 3).

**ITGA7 promotes metastasis by inducing EMT.** As high frequency of ITGA7$^+$ cells was clinically associated with tumour invasion and metastasis, the effect of ITGA7 on metastasis was further studied. Cell migration and invasion assays showed that ITGA7$^+$ cells could significantly promote cell migration and invasion abilities, as compared with ITGA7$^-$ cells (Fig. 5a; Supplementary Fig. 4a). To investigate the effect of ITGA7 on cell mobility, wound healing assay was performed in ITGA7 up-regulated cells and down-regulated cells. The result showed that ITGA7 could markedly enhance tumour cell mobility (Supplementary Fig. 4b). Cell migration and invasion assays also displayed that *ITGA7*-transfected cells could promote cell migration and invasion abilities, as compared with control cells

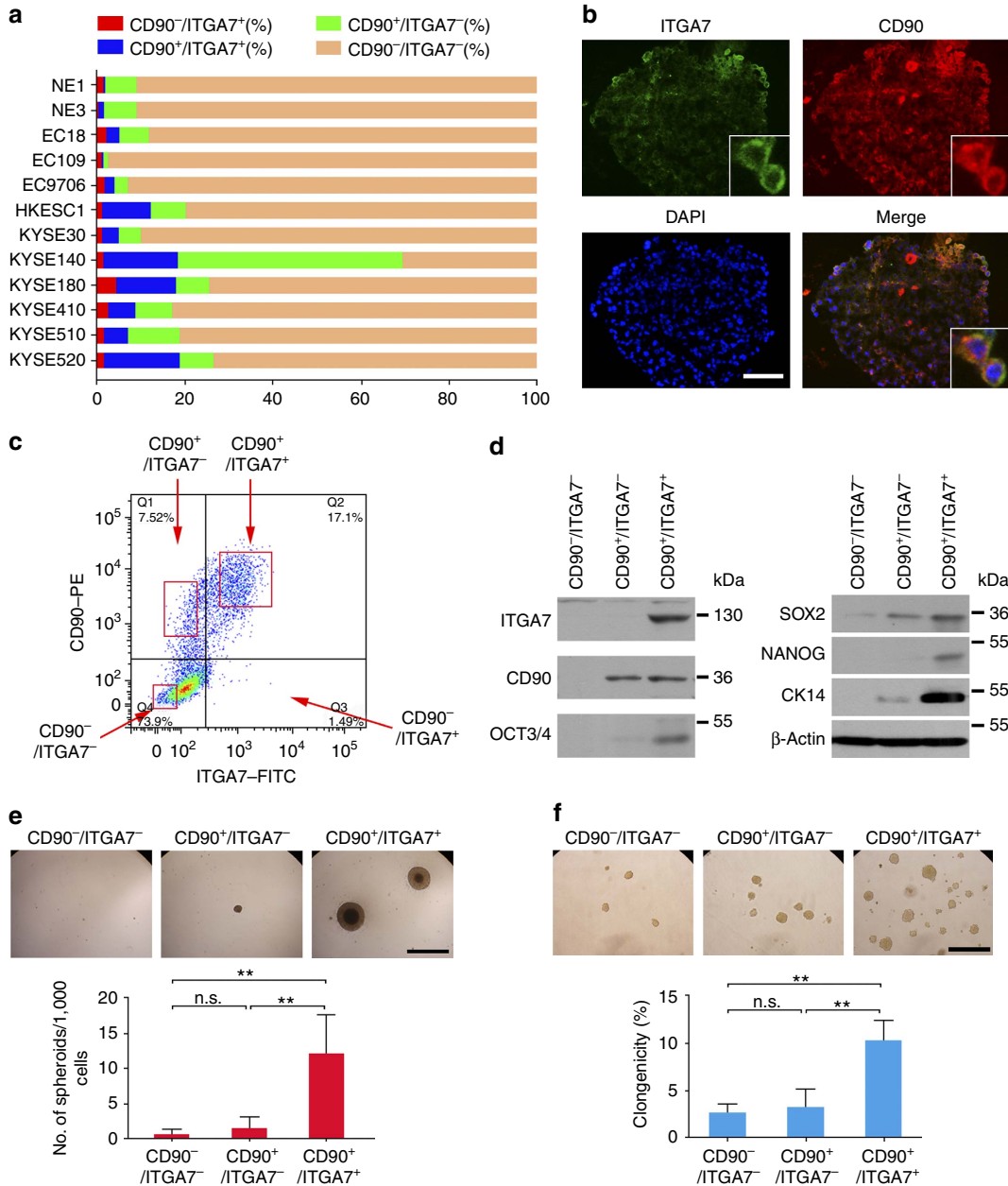

**Figure 2 | ITGA7 and CD90 is co-expressed in OSCC and marks a more tumorigenic OSCC subpopulation.** (**a**) Summary of the overlap of ITGA7$^+$ and CD90$^+$ cells in esophageal cell lines detected by dual-colour flow cytometry. (**b**) Representative dual-colour immunofluorescence analyses of cryosectioned spheroids generated from KYSE520 showing the co-localization of ITGA7 (green) and CD90 (red). Scale bar, 50 μm. (**c**) Representative dual-colour flow cytometry analysis of KYSE520 cells for ITGA7 and CD90. The cells (dot plots) were divided into four quadrants as indicated and collected for each quadrant as gated in the red boxes. CD90$^-$/ITGA7$^+$ subpopulation does not exist because most ITGA7$^+$ cells are also CD90$^+$. Freshly isolated subpopulations were detected by western blotting analysis (**d**), and subjected to spheroid formation assay (**e**, Scale bar, 500 μm) and soft agar colony formation assay (**f**, Scale bar, 500 μm). Data are represented as mean ± s.d. of three separate experiments. Statistics: (**e,f**) ANOVA with *post hoc* test. **P<0.001; n.s., P≥0.05.

(Supplementary Fig. 4c). Once endogenous *ITGA7* was silenced by shRNA, the abilities of cell migration and invasion were significantly inhibited (Fig. 5b). To further investigate the effect of ITGA7 on tumour metastasis, a lymph node metastasis animal model was performed by injecting *ITGA7*-transfected KYSE30 cells into right hind foot-pad of NOD/SCID mice. Empty vector-transfected cells were used as controls. In the lymph node metastasis model, popliteal lymph nodes, which represent the sentinel lymph node for the model, were examined[22]. Swollen popliteal lymph nodes were observed in all seven mice injected with *ITGA7*-transfected cells, and lymph node metastasis was

confirmed by H&E staining in all seven mice. Only one-seventh of mice injected with control cells displayed tumour metastasis (Fig. 5c).

Morphological change from round shape to spindle-like shape was observed in *ITGA7*-transfected cells, suggesting that these cells may undergo epithelial–mesenchymal transition (EMT) (Fig. 5d). To investigate whether this morphological change was caused by EMT, immunofluorescence was performed to test some representative epithelial (E-cadherin and β-catenin) and mesenchymal (fibronectin and vimentin) markers. Compared with control cells, expression of E-cadherin was decreased in

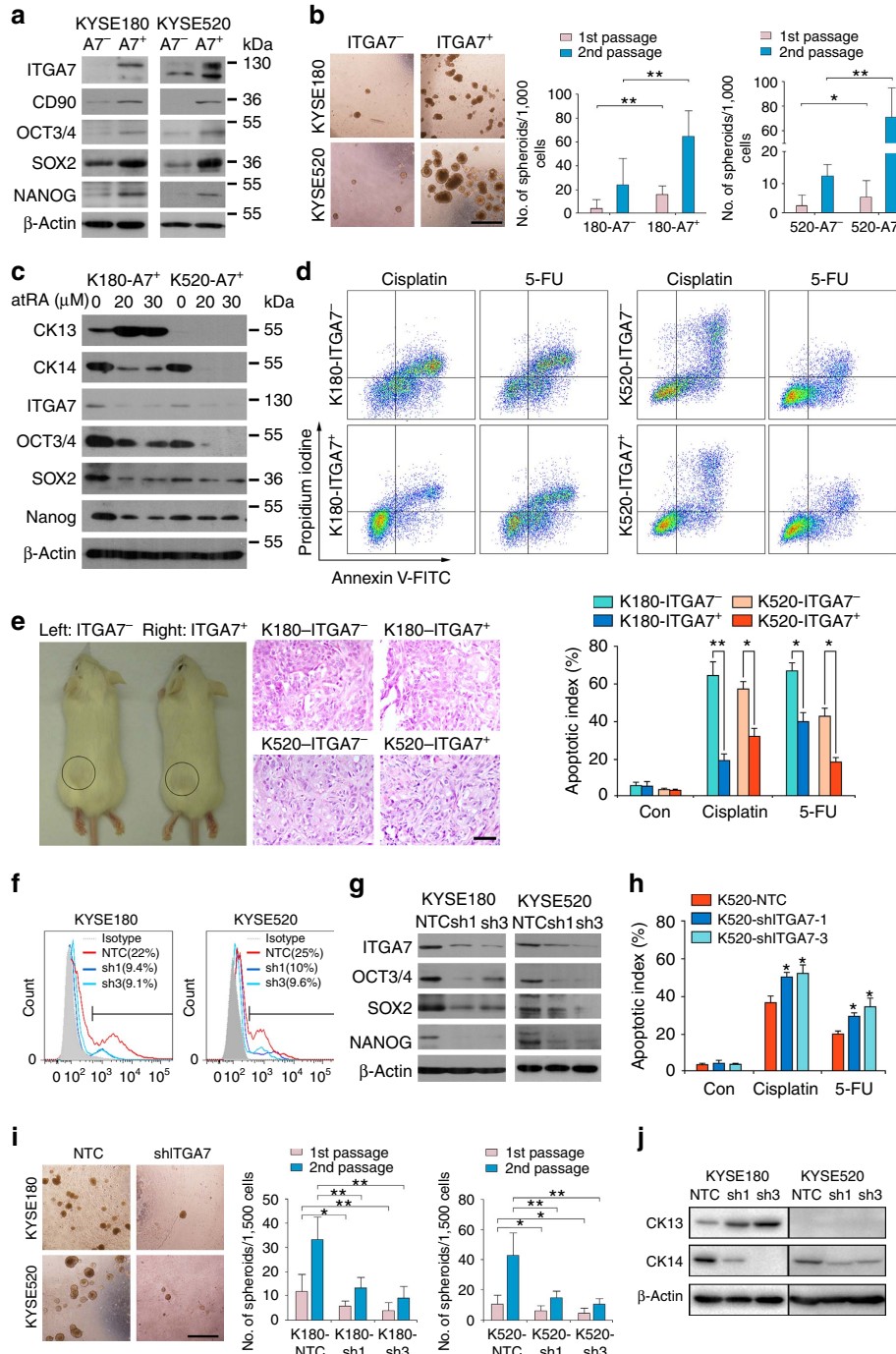

**Figure 3 | ITGA7$^+$ OSCC cells possess the CSC properties, while knockdown of *ITGA7* reduces CSC properties.** (**a**) FACS sorted ITGA7$^+$ cells from KYSE180 and KYSE520 showed higher expression level of stemness-associated genes and CD90, compared with ITGA7$^-$ cells. β-Actin was used as a loading control. (**b**) By spheroid formation assay, the self-renewal ability was enhanced in ITGA7$^+$ cells. Secondary spheroids formed from dissociating spheroids also demonstrated enhanced serial spheroid-forming capacity. The values indicate the mean ± s.d. of two separate experiments with 24 wells per condition. Scale bar, 500 μm. (**c**) On induced by differentiation drug atRA for 5 days, expression of CK13 and CK14 was increased (no CK13 expression in KYSE520) and decreased in ITGA7$^+$ cells, respectively. Decreased expression of ITGA7 and stemness-associated genes was also detected in ITGA7$^+$ cells. (**d**) Representative of apoptosis analyses showing that ITGA7$^+$ cells were more resistant to chemotherapeutic drugs, compared with ITGA7$^-$ cells. The apoptotic index was summarized in the bar chart and the value indicated the mean ± s.d. of three independent experiments. (**e**) Representative of tumours formed in NOD/SCID mice injected with the indicated cells. H&E staining of tumours confirmed a malignant phenotype. Scale bar, 50 μm. (**f,g**) shRNA-mediated *ITGA7* silencing was confirmed by FACS and western blot analysis. The expression of stemness-associated genes was decreased in sh*ITGA7*-transfected cells, compared with non-target control (NTC). (**h**) The summary of apoptotic index showing that *ITGA7* silencing decreased the chemoresistance of KYSE520 cells. The value indicated the mean ± s.d. of three independent experiments. (**i**) Spheroid formation assay showed that the self-renewal ability was decreased in *ITGA7* silencing cells. The values indicate the mean ± s.d. of two separate experiments with 24 wells per condition. Scale bar, 500 μm. (**j**) Knockdown of *ITGA7* promoted the differentiation of OSCC cells. CK13 and CK14 is a well and poor differentiation marker in OSCC, respectively. Statistics: (**b**, **d**) Student *t*-test. (**h,i**) ANOVA with *post hoc* test. *$P < 0.05$; **$P < 0.001$.

*ITGA7*-KYSE30 cells. Although the expression level of β-catenin was similar, its nuclear accumulation was observed in *ITGA7*-transfected cells. Expression of mesenchymal markers fibronectin and vimentin was elevated in *ITGA7*-KYSE30 cells (Fig. 5d). The immunofluorescence result was further confirmed by western blot analysis. Compared with negative counterparts, the expression level of mesenchymal markers was markedly increased in *ITGA7*-transfected cells and ITGA7$^+$ cells, while the expression of E-cadherin was decreased in *ITGA7*-transfected KYSE30 cells and KYSE520 ITGA7$^+$ cells, no significant difference was observed in *ITGA7*-transfected EC109 cells and KYSE180 ITGA7$^+$ cells; maybe the regulation of E-cadherin is cell line specific (Fig. 5e). Conversely, *ITGA7* knockdown displayed a reverse tread in expression of epithelial and mesenchymal markers (Fig. 5f).

**ITGA7 drives CSC features via the FAK/MAPK/ERK signalling.** As focal adhesion kinase (FAK) is a major downstream target activated by integrins, we next studied whether ITGA7 can activate FAK and the corresponding FAK/MAPK/ERK pathway. Compared with ITGA7$^-$ cells, ITGA7$^+$ cells displayed an enhanced phosphorylation of FAK, Src and c-Raf with a concomitant increase in the downstream phosphorylation of MEK1/2 and ERK (Fig. 6a). To confirm the FAK/MAPK/ERK pathway could be activated by ITGA7, we also tested the expression of activated kinases in this signalling cascade in *ITGA7*-transfected EC109 and KYSE30 cells. The results demonstrated that the FAK/MAPK/ERK pathway could be activated by ITGA7 (Fig. 6b). Consistently, decreased phosphorylation of FAK, Src, c-Raf, MEK1/2 and ERK was observed when *ITGA7* was silenced (Fig. 6c).

To further confirm the effect of ITGA7 on cancer stemness is through the FAK-mediated pathway, we studied whether a specific FAK autophosphorylation inhibitor Y15 (1,2,4,5-benzenetetraamine tetrahydrochloride) could abolish ITGA7-induced tumorigenicity and metastasis. Addition of Y15

effectively reduced FAK activity and ERK phosphorylation in *ITGA7*-transfected KYSE30 cells (Fig. 6d). Functional assays also found that the treatment of Y15 could significantly inhibit spheroid formation (Fig. 6e), cell migration (Fig. 6f) and invasion abilities (Fig. 6g), and down-regulated expression of OCT3/4, SOX2 and NANOG (Fig. 6h), as compared with untreated cells. Moreover, to investigate whether MAPK/ERK is the downstream player of ITGA7, we also treated *ITGA7*-KYSE30 cells with MEK1/2 inhibitor U0126. After treatment, the phosphorylation of MEK1/2 and ERK decreased markedly in *ITGA7*-treasfected cells (Supplementary Fig. 5a). Functional study showed that U0126 treatment significantly suppressed foci formation (Supplementary Fig. 5b), cell migration (Supplementary Fig. 5c) and invasion abilities (Supplementary Fig. 5d), as compared with untreated cells. Taken together, these results suggested that ITGA7 functionally conferred cancer stemness features in OSCC via activation of the FAK/MAPK/ERK pathway.

**Chemotherapy enriches ITGA7$^+$ CSCs in clinical OSCC tumours.** To obtain further evidence in support of our hypothesis that ITGA7$^+$ cells confer chemoresistance in OSCC tumours, we established a chemoresistant OSCC xenograft model by treating KYSE520 subcutaneously developed tumours in NOD/SCID mice with varying doses of cisplatin from 0 to 5 mg kg$^{-1}$. Such treatment resulted in variable tumour inhibition among the xenografts, in a dose-dependent trend. From the dose–response curve, we found that the dosage of 2.5 or 5 mg kg$^{-1}$ cisplatin could effectively shrink the tumour size (Fig.7a). Interestingly, IHC result showed that the frequency of ITGA7$^+$ cells was markedly enriched in cisplatin-treated tumours (Fig. 7b). Further, cisplatin-treated OSCC xenografts also displayed enhanced expression of stemness-associated genes as well as ITGA7 in a dose-dependent manner (Fig. 7c). Consistently, OSCC cells derived from chemoresistant group exhibited markedly enhanced ability to form spheroids, compared with the tumour cells from control group (Fig. 7d).

We next studied whether chemotherapy could enrich ITGA7$^+$ cells in clinical OSCC specimens by IHC staining. To fulfil this purpose, 22 and 28 clinical OSCC samples were collected from patients that had or had not undergone chemotherapy before esophagectomy, respectively. Percentage of ITGA7$^+$ cells (5.64 ± 4.15%) was significantly enriched in tumours treated with chemotherapy, as compared with tumours without (0.64 ± 0.74%) (Fig. 7e). This finding suggests that ITGA7$^+$ cells possess strong chemoresistant ability that causes their enrichment in OSCC tumours after chemotherapy.

**ITGA7 inhibits apoptosis via activating FAK/Akt signalling.** To investigate the underlying mechanism of ITGA7-mediated chemoresistance, we studied FAK downstream signalling pathway PI3K/Akt, a well-known pathway that mediates cell survival.

**Table 2 | Tumour incidence rate induced by ITGA7$^-$ and ITGA7$^+$ OSCC cells.**

| Cells injected | KYSE180 ITGA7$^+$ | KYSE180 ITGA7$^-$ | KYSE520 ITGA7$^+$ | KYSE520 ITGA7$^-$ |
|---|---|---|---|---|
| 5,000 | 1/4 | 0/4 | 0/4 | 0/4 |
| 10,000 | 1/4 | 0/4 | 1/4 | 0/4 |
| 50,000 | 3/4 | 0/4 | 2/4 | 0/4 |
| 100,000 | 4/4 | 1/4 | 4/4 | 0/4 |
| 300,000 | 4/4 | 3/4 | 4/4 | 2/4 |
| Tumour-initiating frequency* | 1/28,733 | 1/242,532 | 1/46,830 | 1/770,323 |
| P value* | | P = 0.0003 | | P < 0.0001 |

*Limiting dilution analysis.

**Table 3 | Tumour incidence rate induced by indicated OSCC cells.**

| Cells injected | KYSE180 NTC versus shA7 | KYSE520 NTC versus shA7 | EC109 EV versus ITGA7 | KYSE30 EV versus ITGA7 |
|---|---|---|---|---|
| 10,000 | 1/4 versus 0/4 | 0/4 versus 0/4 | 0/4 versus 1/4 | 0/4 versus 0/4 |
| 50,000 | 1/4 versus 1/4 | 1/4 versus 0/4 | 0/4 versus 3/4 | 0/4 versus 3/4 |
| 100,000 | 4/4 versus 2/4 | 3/4 versus 1/4 | 1/4 versus 3/4 | 1/4 versus 3/4 |
| 300,000 | 4/4 versus 3/4 | 4/4 versus 2/4 | 3/4 versus 4/4 | 4/4 versus 4/4 |
| Tumour-initiating frequency* | 1/57,411 versus 1/190,259 | 1/95,663 versus 1/485,851 | 1/316,851 versus 1/49,618 | 1/209,502 versus 1/59,441 |
| P value* | P = 0.0429 | P = 0.016 | P = 0.0026 | P = 0.041 |

*Limiting dilution analysis.

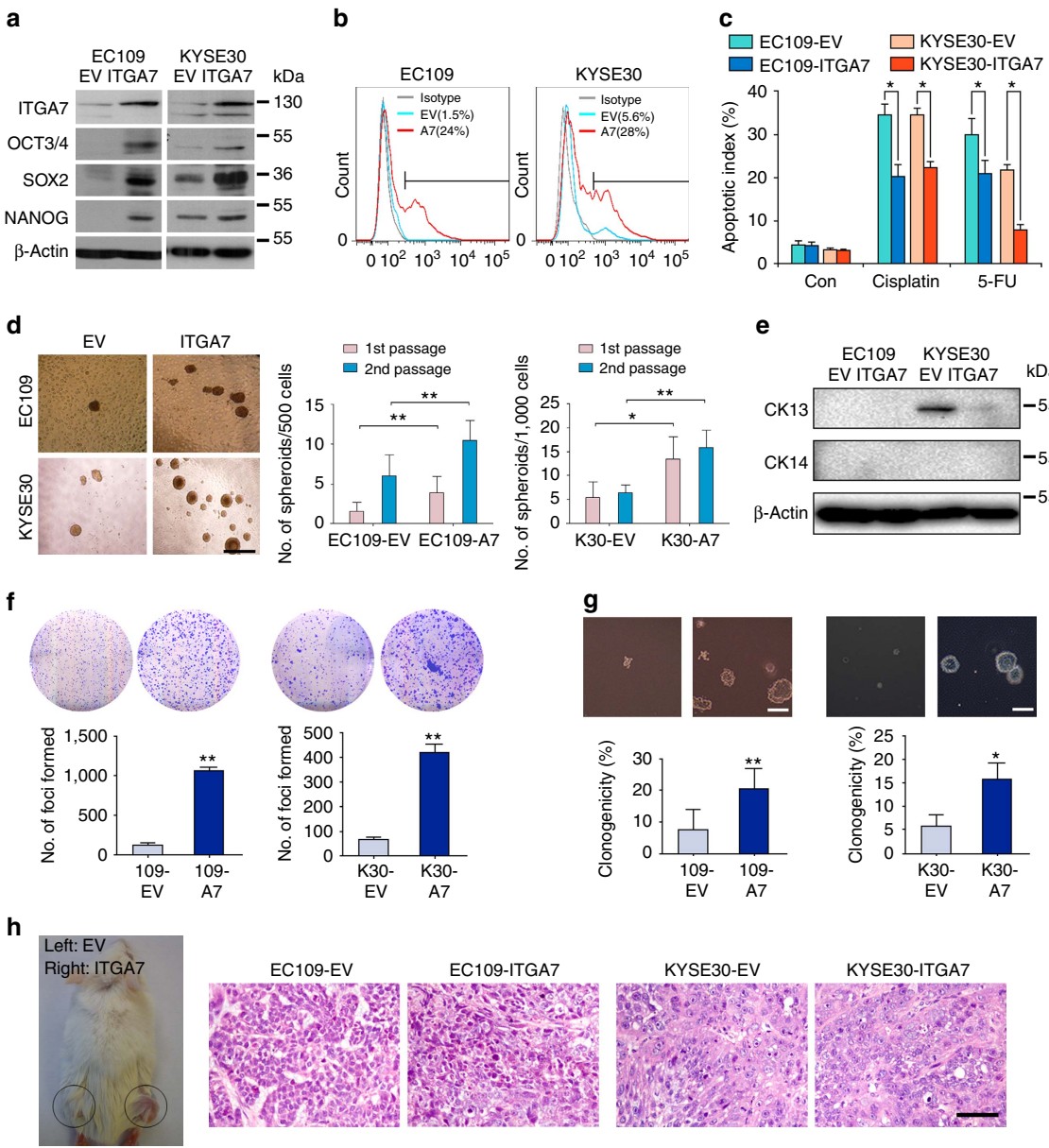

**Figure 4 | Ectopic overexpression of ITGA7 enhances CSC properties in OSCC cells.** (**a**) Expression of ITGA7 and stemness-associated genes was detected by western blotting in *ITGA7*- and empty vector (EV)-transfected cells. β-Actin was used as a loading control. (**b**) The percentage of ITGA7 $^+$ cells was compared by FACS between *ITGA7*- and EV-transfected cells. (**c**) By apoptosis assay, the anti-apoptotic ability of EC109 and KYSE30 was enhanced after *ITGA7* transfection. The apoptotic index was calculated from three independent experiments and presented as mean ± s.d. (**d**) Spheroid formation assay shows the self-renewal ability was enhanced in *ITGA7*-transfected cells. The values indicate the mean ± s.d. of two separate experiments with 24 wells per condition. Scale bar, 500 μm. (**e**) Expression of CK13 and CK14 was detected by western blot analysis in *ITGA7*-transfected and control cells. (**f,g**) Representative images and summaries of foci formation (**f**) and soft agar assays (**g**, Scale bar, 200 μm) performed in *ITGA7*-transfected and control cells. The values indicate the mean ± s.d. of three independent experiments. (**h**) Representative of tumours formed in NOD/SCID mice injected with the indicated cells. H&E staining of tumours confirmed a malignant phenotype. Scale bar, 50 μm. Statistics: (**c,d,f,g**) Student *t*-test. \*$P < 0.05$; \*\*$P < 0.001$.

Activated Akt can phosphorylate a wide variety of substrate proteins to maintain the integrity of the mitochondrial membrane, which blocks cytochrome *c* release and subsequent caspase activation and apoptosis. Therefore, we studied whether ITGA7 could inhibit apoptosis via the FAK/PI3K/Akt signalling. The result found that phosphorylation of FAK and Akt was increased in ITGA7 $^+$ OSCC cells and *ITGA7*-transfected cells on cisplatin stimulation, as compared with control cells (Fig. 7f). As a result, ITGA7 $^+$ cells and *ITGA7*-transfected cells maintained a high inner mitochondrial transmembrane potential ($\Delta\Psi_m$) (shown in red fluorescent), whereas most of the control cells underwent a proapoptotic mitochondrial permeability transition (Fig. 7g). ITGA7-induced Akt phosphorylation protected the mitochondrial membrane from cisplatin-induced collapse, thereby blocking the release of cytochrome *c* into the cytoplasm, and the cleavages of caspase-9, caspase-3 and PARP (Fig. 7f). The anti-apoptotic phenotype and Akt phosphorylation could be effectively reversed when *ITGA7* was silenced in KYSE520 cells. Reduced phosphorylated Akt in *ITGA7* silenced cells led to $\Delta\Psi_m$ collapse (Fig. 7g). As a result, cleaved forms of caspase-9, caspase-3 and PARP increased in *ITGA7* silenced cells compared with control cells (Fig. 7f).

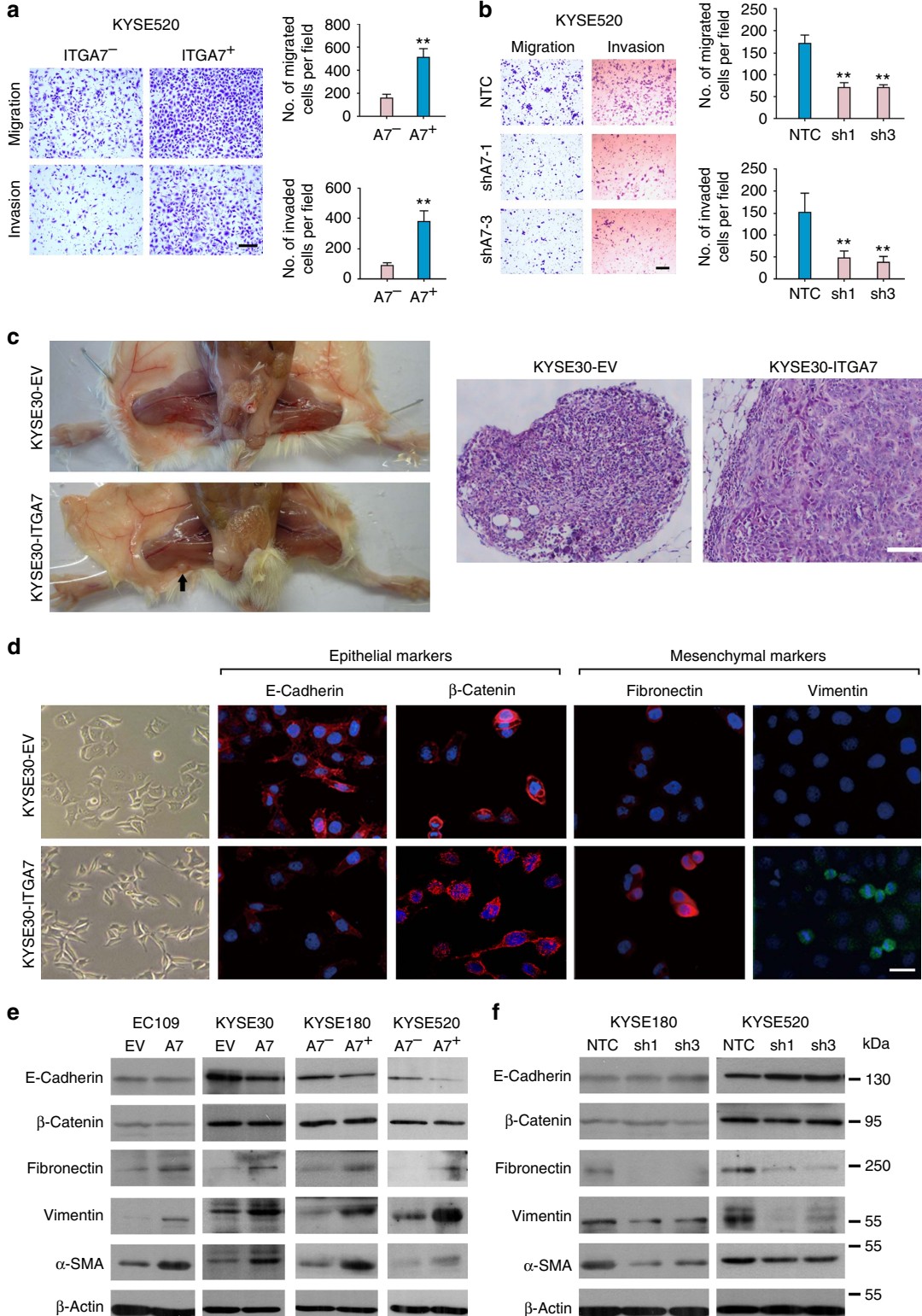

**Figure 5 | ITGA7 promotes metastasis by inducing EMT. (a)** Representatives and summary of migration and invasion assays showing that ITGA7$^+$ cells exhibited enhanced migration and invasion abilities. The values expressed as the mean ± s.d. of three independent experiments. Scale bar, 200 μm. **(b)** Representatives and summary of migration and invasion analyses showing that knockdown of *ITGA7*-suppressed cell motility of KYSE520, compared with their control cells. The values expressed as the mean ± s.d. of three independent experiments. Scale bar, 200 μm. **(c)** Representatives of lymph node metastasis formed in NOD/SCID mice injected with *ITGA7*- or EV-transfected KYSE30 cells. Black arrow indicates the swollen popliteal lymph node. Lymph nodes invaded by tumour cells were confirmed by H&E staining (Scale bar, 100 μm). **(d)** Cell morphology of K30-*ITGA7* and K30-EV cells. Epithelial markers (E-cadherin and β-catenin) and mesenchymal markers (fibronectin and vimentin) were detected by immunofluorescence staining (red: E-cadherin, β-catenin and fibronectin; green: vimentin; blue: nuclear DAPI staining; scale bar, 50 μm). **(e,f)** Expression of EMT markers in indicated cells was detected by western blot analysis. β-Actin was used as a loading control. Statistics: **(a)** Student *t*–test. **(b)** ANOVA with *post hoc* test. **P < 0.001.

## Discussion

So far, the CSCs have been functional defined in a wide variety of solid tumours using extensive markers, including brain, breast, colon, liver and prostate cancers. However, there are several important questions that remain understudied. One question is how sensitive and specific these CSC markers are in marking

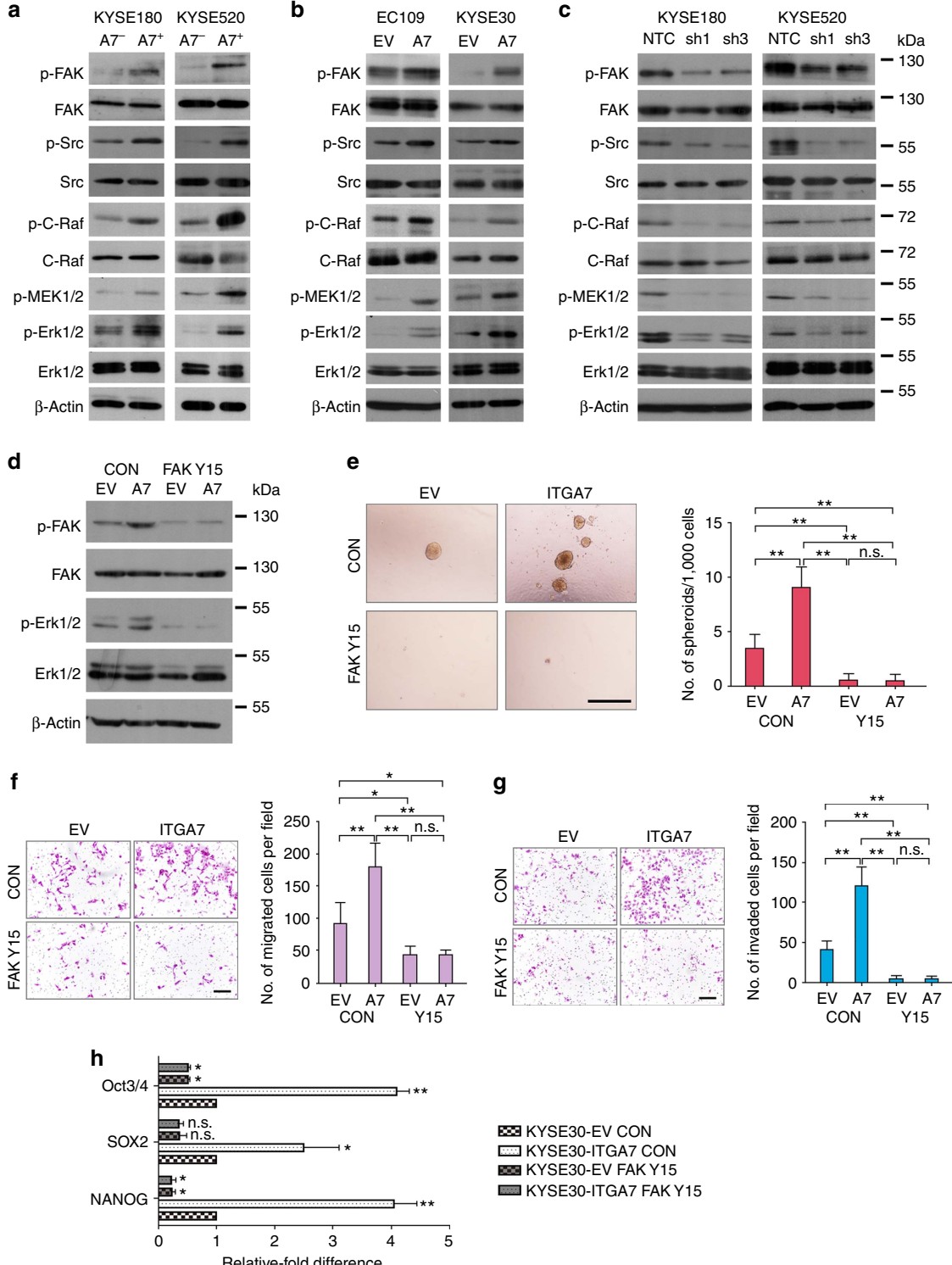

**Figure 6 | ITGA7 drives cancer stemness features in OSCC via the FAK/MAPK/ERK signalling.** (**a–c**) Western blot analyses showed that the FAK/MAPK/ERK pathway was activated in sorted ITGA7[+] cells and *ITGA7*-transfected cells, while inhibited in *ITGA7* silencing cells. β-Actin was used as a loading control. (**d**) Western blot analyses showed that the phosphorylation of FAK and downstream player Erk1/2 was significantly blocked in *ITGA7*-transfected KYSE30 cells after treatment with 7.5 μM FAK inhibitor Y15. Functional assays demonstrated that Y15 treatment could effectively inhibit spheroid formation ability (**e**; 500 μm scale bar), cell migration ability (**f**; 200 μm scale bar), cell invasion ability (**g**; 200 μm scale bar) and the expression of stemness-related genes in *ITGA7*-transfected cells (**h**). The number of spheroids and migrated or invaded cells was calculated from three independent experiments and depicted as the mean ± s.d. in the bar chart. Statistics: (**e–h**) ANOVA with *post hoc* test. \*$P < 0.05$; \*\*$P < 0.001$; n.s., $P \geq 0.05$.

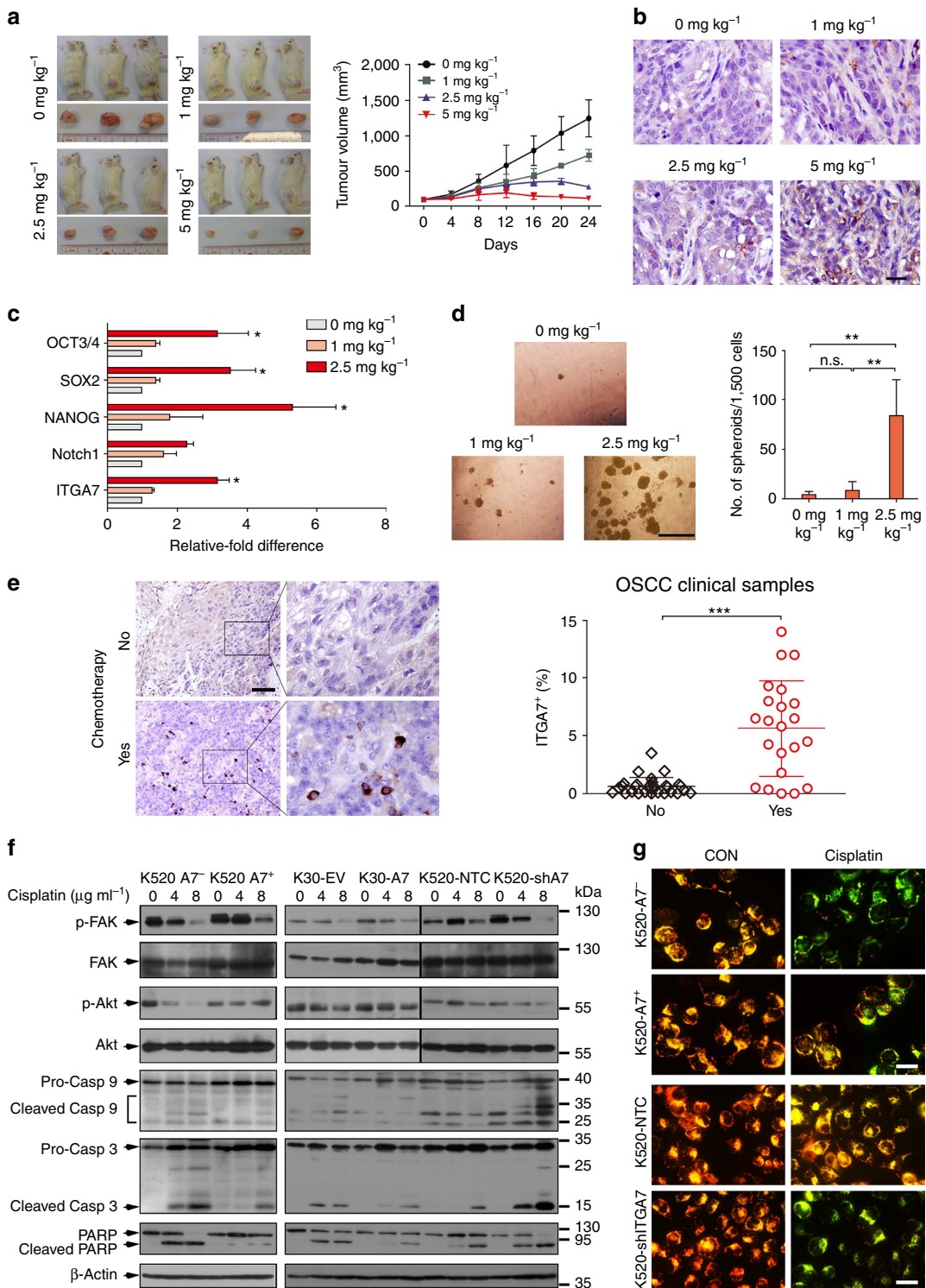

**Figure 7 | ITGA7 promotes chemoresistance via the FAK/Akt/Caspase-9/Caspase-3 pathway.** (**a**) NOD/SCID mice with xenograft tumours of 5 mm in diameter were randomly separated into four groups and treated with varying doses of cisplatin. The doses of 2.5 and 5 mg kg$^{-1}$ cisplatin could effectively suppress tumour growth and shrink tumour size. The average tumour volume was expressed as the mean ± s.d. of three mice. (**b**) Representative IHC images showing that ITGA7$^+$ cells were positively correlated with the dosage of cisplatin. Scale bar, 20 μm. (**c**) qRT-PCR showed that the expression of ITGA7 and stemness-associated genes was positively correlated with the dosage of cisplatin. (**d**) Representatives and summary of spheroid formation assay showing that isolated cells from chemoresistant group displayed enhanced self-renewal ability. Scale bar, 500 μm. (**e**) Representative images of ITGA7 IHC staining of clinical OSCC specimens had or had not undergone chemotherapy treatment. Compared with OSCC specimens without treatment (0.64% ± 0.74%; $n = 28$), ITGA7$^+$ cells were markedly increased after treatment (5.64% ± 4.15%; $n = 22$). Scale bar, 100 μm. (**f**) Western blot analysis was used to detect indicated protein levels in given cells 24 h after cisplatin treatment. (**g**) Representative images of JC-1 dye staining 24 h after cisplatin (4 μg ml$^{-1}$) treatment. JC-1 dye aggregates at high ΔΨm mitochondria (red) or forms monomers at low ΔΨ$_m$ mitochondria (green). Scale bar, 50 μm. Statistics: (**c,d**) ANOVA with *post hoc* test. (**e**) Student *t*-test. *$P < 0.05$; **$P < 0.001$; ***$P < 0.0001$; n.s., $P \geq 0.05$.

CSCs? Another question is the heterogeneity of CSCs in a given cancer. For example, surface markers CD90 (ref. 10), CD44 (ref. 12) and p75[NTR] (ref. 13), have been used for CSCs isolation in OSCC. Does this suggest that different CSC subsets may be derived from different cell of origins, or just in different hierarchical levels? The key issue behind these questions is to identify one specific protein that plays role in regulating stemness and CSC maintenance. Ideally, the optimized marker for CSC should have a function in maintaining CSC in a stem cell state, therefore, the marker should not be lost from the CSC without losing the stem cell properties. As a result, focus has been placed on enzymes or proteins, which can directly associate with CSC maintenance, such as ALDH1 (ref. 23) and ABCB5 (ref. 24). The identification of functional CSC markers has also focused on signalling pathways with an indispensable role in normal stem cell biology, such as Hedgehog[25] and integrin signalling[19–21].

In the present study, we aimed to identify a novel functional CSC marker in OSCC. Non-CG methylation has been confined as a characteristic of stem cells[14,15], and several essential stemness-associated genes, such as NANOG, SOX2 and OCT3/4 have been found highly modified with non-CG methylation in stem cells. It suggests that non-CG methylation may regulate transcriptional activity only in CSCs, but not in non-CSC. Here, we found that non-CG methylation preferentially presented in ITGA7[+] rather than ITGA7[−] OSCC cells, implying that the transcriptional activity of ITGA7 might be also regulated by non-CG methylation. ITGA7 gene encodes for an ECM receptor integrin α7 that belongs to the integrin family. Recent studies indicate that some integrin family members, such as integrin α6 (ref. 19) and β1 (ref. 26), play critical roles in maintaining stem cell niche, preserving a stable stem cell population, and regulating stem cell homoeostasis[27]. In clinical OSCC specimens, ITGA7[+] cells were expressed in <1% of tumour cells in most tumour tissues (203/262, 77.5%), which was consistent with CSC expectation. Interestingly, high-frequency ITGA7[+] cells (>0.6%) was significantly associated with OSCC stemness such as cell differentiation, advanced clinical stage, tumour invasion and lymph node metastasis, suggesting that ITGA7 might play roles in regulating stemness.

Compared with a known oesophageal CSC marker CD90, the population of ITGA7[+] cells was much smaller than the population of CD90[+] cells in both immortalized oesophageal and OSCC cell lines. Interestingly, most of ITGA7[+] cells were co-expressed with CD90, while only part of CD90[+] cells co-expressed with ITGA7 in OSCC cell lines. Functional study found that ITGA7[+]/CD90[+] cells possessed stronger stemness compared with ITGA7[−]/CD90[+] cell, implying that ITGA7[+] cells might be more properly represent CSCs in OSCC. Further functional assays were used to characterize ITGA7[+] cells sorted from KYSE180 and KYSE520 cells. Compared with ITGA7[−] cells, ITGA7[+] cells showed stronger abilities for self-renew, cell differentiation, cell motility and chemoresistance, as well as with higher expression of stemness-associated genes. In addition, ITGA7-transfected cells also possessed CSC properties. We also found ITGA7[+] OSCC cells underwent EMT, which is responsible for initiation of metastasis[28,29]. All these CSC properties could be effectively inhibited when ITGA7 was silenced by shRNA. All these findings suggest that ITGA7 is a better CSC marker with stemness regulatory function. As expression level of ITGA7 was extremely low in clinical specimens, these functional assays could not be repeated in clinical samples. Further study found that the effects of ITGA7 on OSCC development might be through the activation of FAK/MAPK/ERK signalling pathway, which has been reported to be involved in a series of tumour processes[30]. Similar to ITGA7 knockdown, FAK inhibitor could also inhibit the effect of ITGA7 on self-renewal ability and

decrease the expression of stemness-associated genes such as OCT3/4, SOX2 and NANOG. Recently, Golubovskaya[31] reported the interaction of FAK, NANOG, OCT3/4 and SOX2 played a critical role in regulating CSCs. This suggests that ITGA7 might be involved in stemness regulation and CSC maintenance in OSCC through the activation of FAK/MAPK/ERK signalling pathway.

One of the important characteristics of CSCs in clinical point of view is its resistance to conventional chemotherapy. In the present study, we demonstrated that ITGA7[+] cells were markedly enriched in residual OSCC xenograft tumours in NOD/SCID mice after treatment with cisplatin chemotherapy. Consistent with this data, the enrichment of ITGA7[+] cells was also observed in clinical OSCC samples after chemotherapy. This finding may help explain why chemotherapy fails to eradicate the disease although it can effectively shrink the tumour size by killing most of tumour cells. Therefore, development of novel therapeutic strategy by targeting CSCs is necessary to cure cancer. Further study found that ITGA7 exerted the anti-apoptotic effect through the versatile player FAK to activate the Akt signalling, which subsequently inhibited the release of cytochrome c into the cytoplasm and the cleavages of caspase-9, caspase-3 and PARP. In summary, we report here that ITGA7 is a potential CSC marker in OSCC with function in stemness regulation and CSCs maintenance. Further characterization of ITGA7 may provide an alternative therapeutic approach by targeting CSCs to achieve better clinical outcome.

## Methods

**OSCC clinical specimens.** Three hundred pairs of primary OSCC specimens and their adjacent non-tumour oesophageal tissues, used for a TMA, were collected from patients who underwent oesophageal resection at Linzhou Cancer hospital (Henan, China). Twenty-eight clinical specimens were collected from OSCC patients at Sun Yat-Sen University Cancer Center (Guangzhou, China). None of these patients received preoperative treatment. Twenty-two clinical specimens were collected from patients with OSCCs who had received neoadjuvant chemo-radiotherapy before operation at the Queen Mary Hospital (Hong Kong). Histological examination was carried out by pathologists, and diagnosis was made based on the microscopic features of the carcinoma cells. Tumours were graded using the American Joint Committee on Cancer (AJCC)/International Union Against Cancer (UICC) tumour staging system. Informed consent was obtained from all patients before the collection of oesophageal specimens, and the study was approved by the committees for ethical review of research involving human subjects at Zhengzhou University (Zhengzhou, China), Sun Yat-Sen University (Guangzhou, China) and the University of Hong Kong (Hong Kong).

**Cell lines.** Immortalized oesophageal epithelial cell lines NE1 and NE3 were established in Professor George Tsao's laboratory (Department of Anatomy, The University of Hong Kong). Chinese OSCC cell lines HKESC1, EC18, EC109 and EC9706 were kindly provided by Professor Srivastava (Department of Pathology, The University of Hong Kong). Six Japanese OSCC cell lines (KYSE30, KYSE140, KYSE180, KYSE410, KYSE510 and KYSE520) were obtained from DAMZ, the German Resource Center for Biological Material. Mycoplasma contamination was not found in these cell lines.

**Animals.** The study protocol was approved by and performed in accordance with the Committee of the Use of Live Animals in Teaching and Research at the University of Hong Kong. For in vivo tumorigenic experiment, various numbers of ITGA7 sorted cells, ITGA7-transfected cells and ITGA7-suppressed cells were injected subcutaneously into 4-to-5-week-old NOD/SCID mice. Mice were killed between 2 and 4 months post injection. Those animals injected with tumour cells but no sign of tumour burden were generally terminated 4 months after tumour cell inoculation, and mice were opened up at the injection sites to confirm that there was no tumour development. For in vivo metastasis assay, each experimental group consisted of seven 5-week-old NOD/SCID mice. Briefly, $2 \times 10^5$ cells in 20 µl phosphate-buffered saline (PBS) were injected into the right hind foot-pad of each mouse. All of the mice were euthanized 4 weeks after injection. The right side popliteal lymph nodes were excised and embedded in paraffin. For chemoresistant xenograft model, subcutaneous xenografts in 5-week-old NOD/SCID mice were established with KYSE520 cells ($1 \times 10^6$). Treatment was started once the size of the xenograft reached ∼5 mm in diameter. The mice were randomly assigned into four groups, each consisting of three mice. They were treated with cisplatin intraperitoneally every 4 days for 24 days at different doses of 0, 1, 2.5 or

$5\,\mathrm{mg\,kg^{-1}}$. Tumour sizes were measured by caliper, and tumour volumes were calculated as volume $(\mathrm{mm^3}) = L \times W^2 \times 0.5$. Cell dissociation from xenograft is detailed in Supplementary Methods.

**IHC.** IHC staining was performed using the standard streptavidin–biotin–peroxidase complex method. Briefly, paraffin sections were deparaffinized and rehydrated. Slides were heated for antigen retrieval for 15 min in 10 mM citrate (pH 6). Sections were incubated with monoclonal mouse anti-human ITGA7 (1:100 dilution; Abgent) at 4 °C overnight. EnVision Plus System-HRP (DAB; DAKO) was used according to manufacturer's instruction, and followed by Mayer's hematoxylin counterstaining. Stained slides were imaged on an AperioScanscope CS imager (Vista, CA, USA).

**Immunofluorescence.** Immunofluorescence was performed on cell lines and cryosectioned spheroids. First, cells on the coverslips were fixed with 4% paraformaldehyde, and incubated with primary antibody (rabbit anti-ITGA7 or mouse anti-CD90) overnight at 4 °C. After washing, cells underwent incubation with FITC- or PE-conjugated secondary antibodies for 1 h, and subsequently counterstained with DAPI for 5 min at room temperature (Roche Diagnostics). All images were visualized under Leica DMRA fluorescence microscope (Wetzlar, Germany).

**Flow cytometry and cell sorting.** Flow cytometric analysis and cell sorting was conducted using anti-human ITGA7 (Abgent), and PE-conjugated mouse anti-human CD90 (BD Biosciences). Cells were gently detached in citric saline buffer, and incubated in PBS containing 2% fetal bovine serum (FBS) with either PE-conjugated primary antibody or primary antibody followed by a FITC-conjugated secondary antibody. Isotype-matched mouse or rabbit immunoglobulin served as controls to gate positive cells. Samples were analyzed and sorted on BD LSR Fortessa Analyzer and FACSAria I Cell Sorter (BD Biosciences), and data analyzed by FlowJo software (Tree Star). For ITGA7 cell sorting, only the top 10% most brightly stained or the bottom 15% most dimly stained cells were gated as positive and negative populations, respectively. Cell viability was assessed using trypan blue exclusion. Using antibodies listed in Supplementary Table 4.

**Western blot analysis.** Quantified protein lysates were resolved on SDS–PAGE, transferred onto a polyvinylidenedifluoride (PVDF) membrane (Millipore), and then blocked with 5% non-fat milk in Tris-buffered saline-Tween 20 (TBS-T) for 1 h at room temperature. The blocked membrane was then incubated with primary antibody at 4 °C overnight. After washing with TBS-T, the membrane was incubated for 1 h with horseradish peroxidase (HRP)-conjugated secondary antibody. A complex of primary and secondary antibodies-labelled proteins were detected by enhanced chemiluminescence (ECL) system (GE Healthcare) and X-ray film (GE Healthcare). β-actin was used as a housekeeping control. Using antibodies listed in Supplementary Table 4. Full blots of Figs 3c and 7f are provided in Supplementary Fig. 6.

**DNA extraction and non-CG methylation analysis.** Genomic DNA was extracted from isolated $\mathrm{ITGA7^+}$ and $\mathrm{ITGA7^-}$ cells by phenol–chloroform method followed by bisulfite modification using the EpiTECT Bisulfite Kit (Qiagen). BGS primers were listed in Supplementary Table 5. The template was amplified by PCR for 35 cycles and then cloned into the pGEM-T Easy vector (Promega) and sequenced as individual clones by Beijing Genomics Institute. The sequence data was analyzed by BiQ analyzer (Max Planck Institute).

**RNA extraction and quantitative real-time PCR.** Total RNA was extracted using TRIZOL Reagent (Invitrogen), and cDNA was synthesized using a reverse transcription (RT)-PCR Kit (Roche) according to the manufacturer's instructions. Quantitative real-time PCR (qRT-PCR) was performed using the SYBR Green PCR Kit (Applied Biosystems) and an ABI PRISM 7900 Sequence Detector (Applied Biosystems). Specificity of primers was verified by dissociation curve analysis. Data was analyzed using ABI SDS v2.4 software (Applied Biosystems). All qRT-PCR reactions were performed in duplicates. Housekeeping gene GAPDH was used as an internal control. Using primers listed in Supplementary Table 5.

**Spheroid formation assay.** Single cells were cultured in suspension in 24-well polyHEMA (Sigma-Aldrich)-coated plates. Cells were grown in 300 μl of serum-free DMEM/F12 medium (Invitrogen) supplemented with $4\,\mathrm{\mu g\,ml^{-1}}$ insulin (Sigma-Aldrich), B27 (1:50; Invitrogen), $20\,\mathrm{ng\,ml^{-1}}$ human recombinant EGF (Sigma-Aldrich), $10\,\mathrm{ng\,ml^{-1}}$ human recombinant basic FGF (Invitrogen). Cells were replenished with 30 μl of supplemented medium every second day. For serial passage of primary spheroids, the primary spheroids were collected and dissociated to single cells using TrypLE Expression (Invitrogen). Following dissociation, trypsin inhibitor (Invitrogen) was used to neutralize the reaction, and cells were resuspended in DMEM/F12 medium with the above supplements.

**Apoptosis assay.** Cells were treated with cisplatin or 5-FU (KYSE180 with $6\,\mathrm{\mu g\,ml^{-1}}$ cisplatin or $12\,\mathrm{\mu g\,ml^{-1}}$ 5-FU; KYSE520 with $10\,\mathrm{\mu g\,ml^{-1}}$ cisplatin or $30\,\mathrm{\mu g\,ml^{-1}}$ 5-FU; EC109 with $4\,\mathrm{\mu g\,ml^{-1}}$ cisplatin or $20\,\mathrm{\mu g\,ml^{-1}}$ 5-FU; and KYSE30 with $1\,\mathrm{\mu g\,ml^{-1}}$ cisplatin or $5\,\mathrm{\mu g\,ml^{-1}}$ 5-FU) for 48 h. Cells then were collected and stained with propidium iodide (PI), and FITC-conjugated Annexin-V provided by the Annexin-V FLUOS Staining Kit (Roche Diagnostics) according to manufacturer's instructions. Analysis was determined by flow cytometry on a LSR Fortessa Analyzer (BD Biosciences) and FlowJo software (Tree Star).

**Differentiation assay.** atRA (listed in Supplementary Table 6) was used to treat sorted $\mathrm{ITGA7^-}$ and $\mathrm{ITGA7^+}$ cells at a concentration of 20 or 30 μM in RPMI 1640 medium supplemented with 10% FBS and 1% penicillin and streptomycin for 5 days, with medium change daily. Expression change of ITGA7, differentiation markers CK13 and CK14, and stemness-associated genes, was assessed by western blot analysis.

**In vitro tumorigenic assays.** Anchorage-dependent growth was assessed by foci formation assay. Briefly, $2 \times 10^3$ cells were seeded in a 6-well plate. Surviving colonies (> 50 cells per colony) were stained with 1% crystal violet (Sigma-Aldrich) and counted after 1 week. Anchorage-independent growth was assessed by colony formation in soft agar. In brief, $5 \times 10^3$ cells mixed thoroughly in 0.35% low melting soft agar (BD Biosciences) were grown in a 6-well plate precoated with 0.5% agar. After 2 weeks, colonies ($\geq 10$ cells) were counted under microscope in 10 fields per well and photographed.

**ITGA7 overexpression and knockdown.** ITGA7 overexpression plasmid was kindly provided by Dr. Jianhua Luo (Department of Pathology, University of Pittsburgh School of Medicine). Briefly, the full-length wild-type ITGA7 cDNA (NM_002206.2) was cloned into the pLenti6 expression vector (Invitrogen) and transfected into EC109 and KYSE30 cells with Lipofectamine 2000 Reagent (Invitrogen). Stable ITGA7-expressing clones were selected for 2 weeks with blasticidin (Sigma-Aldrich). EV-transfected cells were used as controls. For ITGA7 knockdown, the scrambled shRNA plasmid (pLKO.1-NTC) and the ITGA7-specific shRNA expression vectors (pLKO.1-shITGA7) were purchased from Sigma-Aldrich. Two constructs against ITGA7 were used: shITGA7-1 (TRCN0000057708) and shITGA7-3 (TRCN0000057711). The shRNA sequences listed in Supplementary Table 7. The pLKO.1-shITGA7 or the scrambled shRNA plasmid was transfected into KYSE180 and KYSE520, and stable clones were selected using puromycin (Sigma-Aldrich).

**Migration and invasion assays.** Migration and invasion assays were performed in 24-well millicell hanging insert (Millipore) or 24-well BioCoat Matrigel Invasion Chambers (BD Biosciences). In brief, $1 \times 10^5$ cells were seeded to the top chamber and 10% FBS in medium was added to the bottom chamber as a chemoattractant. After 24 or 48-h incubation, the number of cells that invaded through the membrane (migration) or Matrigel (invasion) was counted in 10 fields and imaged using SPOT imaging software (Nikon).

**Mitochondrial membrane potential assay.** The mitochondrial membrane potential $(\Delta\Psi_\mathrm{m})$ was detected with a MitoPTJC-1 detection kit (Immunochemistry Technologies) according to the manufacturer's protocol. In brief, cells were seeded onto coverslips and cultured to 80% confluence; then $4\,\mathrm{\mu g\,ml^{-1}}$ cisplatin was added to treat cells for 24 h. The cells were washed twice with PBS and incubated with the $\Delta\Psi_\mathrm{m}$-sensitive dye JC-1 at 37 °C for 15 min. Images were visualized immediately under Leica DMRA fluorescence microscope (Wetzlar, Germany).

**Statistics.** Statistical analysis was carried out with SPSS v. 16 (Chicago, IL, USA). Pearson $\chi^2$ test was used to analyze the association of ITGA7 expression with clinicopathologic parameters. Kaplan–Meier plot and log-rank test were used for survival analysis. Independent Student t-test and ANOVA with post hoc test were used for most studies as indicated in the figure legends. For limiting dilution assay, the tumour-initiating frequency and statistics were calculated using ELDA software[32]. The data are presented as the mean ± s.d. of three independent experiments. The P-values were denoted as $*P < 0.05$, $**P < 0.001$ and $***P < 0.0001$ in all figures.

**Data availability.** The data that support the findings of this study are available from the corresponding author upon request.

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

## Acknowledgements

We thank the Faculty Core Facility of the LKS Faculty of Medicine at The University of Hong Kong for help with the flow cytometry cell sorting, as well as Dr. Jianhua Luo (Department of Pathology, University of Pittsburgh School of Medicine) for providing the *ITGA7*-expressing plasmid. This work was supported by grants from the Hong Kong Research Grant Council (RGC) grants including Colla-borative Research Funds (C7038-14G and C7027-14G), General Research Funds (HKU/7668/11M, 767313 and CUHK/766613), NSFC/RGC Joint Research Scheme (N_HKU712/12), Hong Kong Health and Medical Research Found (02133366), grants from the National Basic Research Program of China (2012CB967001), China National Key Sci-Tech Special Project of Infectious Diseases (2013ZX10002-011-005), National Natural Science Foundation of China (81272416 and 81372583) and the Science and Technology Foundation of Shenzhen (CXZZ20150430092951135 and KQTD20140630100658078). Professor X.Y. Guan is Sophie YM Chan Professor in Cancer Research.

## Author contributions

Conceptualization: X.-Y.M., L.F. and X.-Y.G.; methodology: X.-Y.M., L-Y.Z. and X.-Y.G.; investigation: X.-Y.M., L-Y.Z. and T-T.C.; writing–original drafts: X.-Y.M.; Writing—review & editing: X.-Y.M., S.M. and X.-Y.G.; funding acquisition: L.F. and X.-Y.G.; resources: Y-R.Q., K.W.C., S.M. and D.X. and supervision: X.-Y.G.

## Additional information

**Competing financial interests:** The authors declare no competing financial interests.

