## [Peer Review File · Nature Communications]

Reviewers' comments:

Reviewer #1 (Remarks to the Author):

The authors identify integrin Alpha 7 (ITGA7) as a putative CSC marker in esophageal squamous cell carcinoma. Previously, this group had identified CD 90 as a putative cancer stem cell marker and they now present evidence that a sub-fraction of CD 90 expressing cells also express ITGA7 and that these cells are enriched for CSC properties. Furthermore, they present evidence suggesting that ITGA7 regulates CSC properties via activation of focal adhesion kinase (FAK) and downstream signaling pathways including ERK, MAPK and AKT. In general, the experiments are well done and reported providing evidence for a functional role of ITGA7 in the stem cell phenotype of esophageal squamous cell carcinoma cells.

However, a number of issues need to be addressed:

1. The authors initially identified ITGA7 as a putative CSC marker by virtue of its higher non GC methylation frequency. Based on this they speculate that ITGA7 might be activated by a non GC methylation (figure 1E). However, they only provide correlative evidence and there is no direct evidence for regulation of ITGA7 by non GC methylation in esophageal cancer stem cells. The authors either need to perform functional expression studies or acknowledge that their data are only correlative.
2. The most definitive assay for demonstration of tumor initiating capacity is the ability to form tumors in immunosuppressed mice. The authors do provide such data in figure 3B demonstrating the increased tumor initiating capacity of ITGA7+ compared to ITGA7- cells. The authors need to calculate tumor initiating frequency utilizing extreme limiting dilution analysis. In addition, rather than just examining ITGA7+ versus ITGA7- cells the authors should directly compare the tumor initiating frequency of ITGA7+ cells to CD 90+ ITGA7- cells rather than all ITGA7- cells. This is important since the authors claim that ITGA7 is a more specific marker of CSC's than CD 90. The manuscript could be further strengthened if similar *in vivo* experiments were performed on ITGA7 knockdown cells or ITGA7 overexpressing cells. Although Figure 4H, which shows a higher growth rate and larger median tumor volume in tumors overexpressing ITGA7 is interesting, these characteristics are more reflective of growth rate of transient amplifying cells than of actual tumor initiating capacity. As was done in Figure 3.E the authors need to perform tumor initiating frequency studies.
3. The authors demonstrate that ITGA7 knockdown reduces FAK activation. Other investigators have demonstrated that Beta-1 integrins are crucial regulators of FAK phosphorylation. The other tumor types alpha-6 beta-1 which is also a laminin receptor activates focal adhesion kinase through interactions with beta-1 integrin. Have the authors performed any experiments to elucidate whether hetero-dimerization of ITGA7 to beta-1 integrin is necessary to activate focal adhesion kinase?
4. The authors provide convincing evidence that ITGA7 activates focal adhesion kinase and that a specific FAK auto phosphorylation inhibitor Y15 abolishes ITGA7 induced tumorigenesis and metastasis. Although they present western blotting showing that FAK activation results in downstream activation of MAPK and ERK, they provide no functional evidence that this downstream pathway mediates ITGA7 biological activities. The authors need to perform experiments utilizing MAPK and/or ERK inhibitors to show the importance of this pathway by demonstrating that these inhibitors abrogate the effect of ITGA7 overexpression.
5. The authors compare ITGA7 expression in clinical ESCC specimens of patients who either have received chemotherapy before esophagectomy or had esophagectomy in the absence of chemotherapy and demonstrate that patients with preoperative chemotherapy had significantly higher percentage of ITGA7+ cells. While these results are interesting they can potentially be biased by patient selection as to which patients were eligible to receive preoperative chemotherapy. Is it possible that patients were selected to receive preoperative chemotherapy based on their more aggressive disease and that this could account for the increase in ITGA6 expressing cells? The authors should comment on potential biases introduced by patient selection.

Reviewer #2 (Remarks to the Author):

The manuscript "Integrin $\alpha 7$ is a functional cancer stem cell marker in esophageal squamous cell carcinoma" by Ming et al., describes a role for Integrin $\alpha 7$ in cancer progression. The authors propose that a subpopulation of ITGA7+ cells in ESCC correlates with aggressive tumor behavior, poor outcome and cancer stemness and thus suggest ITGA7 is a CSC marker. Although it has been shown in a broad range of cancers that cell surface integrins including ITGA7 activate signaling pathways which lead to cancer stemness. In particular, ITGA7 has been linked to stemness of glioblastoma and mesenchymal cells. Also, the integrin signaling pathway they describe has already been linked to cancer aggressiveness and stemness in previous published work. To prove that ITGA7 expressing cells have cancer stem cell properties, the authors use ESCC cell lines and various stem cell assays; however the results do appear somewhat preliminary and not particularly unexpected based on previous reports

Specific comments:

1) The major weakness of the paper is the novelty, as ITGA7 has been associated with stem cells in other tissues (mesenchymal) and cancers (glioblastoma) and this integrin signaling pathway is established and has been shown to result in a more metastatic and aggressive cancer. However, this is the first report linking ITGA7 as a biomarker for ESCC stemness. This is based on the evaluation of cultured cell lines. To make their findings more relevant to the clinical course of ESCC the authors should examine patient derived xenograft (PDX) model for tumor initiation and the link to ITGA7. At least this approach would improve the relevance of their findings in terms of tumor cell stemness and cancer progression.

2) The author's aim is to demonstrate that ITGA7+ ESCC cells possess CSC properties. Using cell lines, they perform stem cell assays both in vitro and in vivo to determine ITGA7+ stem properties. Using FACS sorting they show ESCC cell lines express ITGA7 in a small sub population of cells. These cells also co-express cancer stem marker CD90 (previously published by the group) as shown in multiple panels in Figure 1. In order to prove ITGA7 expression is involved in cancer stemness they performed a knockdown of ITGA7 using shRNA (shITGA7). However the comparison should not be between control and shITGA7 of the entire cell line population since the majority of the naïve cells are already ITGA7- (Figure 3h,i and Figure 5b). The experiment should be done by analyzing the CD90+ subpopulation with and without ITGA7 knockdown to show a differential effect on CSC properties. Similar sorting for CD90 positive cells should be done prior to western blot analysis using ITGA7 knockdown (Figure 3g,j and Figure 5E). The knockdown was only partial with a 50% decrease in expression of an already small population (22-25%). So it would seem analysis of ITGA7 knockdown in the subpopulation of CD90 expressing cells, which the authors claim co-express ITGA7, would be a better approach.

3) Figure 1B the authors do not describe where the data for the KM plot was obtained from. Is this from the 262 primary ESCC cases (Table 1)? If so do they use the same cut off value (0.6%) for high and low expression to represent ITGA7- and ITGA7+ in the survival plot? Please clarify in results, figure legend or methods section.

4) In figure 1E, the authors show non cg-methylated sites in the ITGA7 gene loci of ITGA7+ cells, implying that ITGA7 expression might be activated by non-CG-methylation. Although interesting, other than this one piece of data (Figure 1E) they do not further elaborate or show the relevance of non-CG-methylation in ESCC tumor aggressiveness, stemness, overall patient outcome or ITGA7 expression. To strengthen this claim, further analysis is required. For example, studies

could be performed using DNA methylase inhibitors to show an effect on ITGA7 expression. Alternatively, they should show a correlation between poor overall survival and non-methylated ITGA7 in ESCC patient tumors. Otherwise the data provided are not particularly relevant to the rest of the study.

Reviewer #3 (Remarks to the Author):

In the manuscript by Ming et al, the authors identify integrin $\alpha 7$ (ITGA7) as a novel marker of a cancer cell population with stem-like properties. They further go on to show that this population of cells is more resistant to chemotherapy (5FU and cisplatin), and that they exhibit increased migration and invasion. Finally, they propose that the mechanism through which ITGA7 regulates these cellular properties is through activation of FAK/MEK/ERK signalling.

I think that aspects of this study are interesting. However, I feel that the authors have not gone far enough in addressing the pertinent questions arising from the work they present. When finished reading the manuscript I am left wondering whether ITGA7 is merely a marker of cancer stem cells (CSC) or is it functionally relevant in the regulation of the CSC phenotype. The link between integrins, FAK etc and chemoresistance, increased migration and invasion is in no way surprising or indeed novel. Therefore, I believe the authors need to show more definitively that ITGA7 not only marks CSCs but also plays an important role in regulating their phenotype and stem-like properties, and that FAK plays an important role downstream. The properties of these cells in terms of chemoresistance, invasion etc should come after this to describe the importance of these cells in terms of therapy.

My specific concerns are as follows:

Figure 1:

1. The immuno-staining in the top left panel of figure 1A does not look to have worked very well (large white area). Either this is a poor choice of image or the staining quality is poor and should be repeated.
2. In the text it is stated that based on frequency of ITGA7+ cells in the staining the samples have been divided up into high and low frequency group. This is based on what seems like a somewhat arbitrary cutoff of 0.6%. Is there a valid reason for this cutoff? If so, this should be stated in the text. Otherwise, what led to this choice of cutoff?
3. In figure 1C I would like to see a plot for each of the cell lines (although this could be supplementary) and statistics comparing the data for the two normal cell lines to the 10 cancer lines. Also, immunofluorescence data presented in d does not really support the frequencies presented in c, and I am not sure how useful this data really is?

Figure 2:

1. In figure 2a the gating used is different to that used in the other cell lines and there is no valid reason provided for this. Also, I want to see a scatter plot with gating for every cell line plus controls. When looking at the NE1 scatterplot it appears that there are two populations, a CD90- and a CD90+, and these would be apparent if the same gating had been used. It looks like there is a CD90+ population in the NE1 and that this population may acquire ITGA7 expression in the cancer cells. This would change the interpretation of the authors results presented in B. The authors should provide all of the data on which these conclusions are based, including controls, and if the gating is significantly different justify why. Otherwise the gating should be changed and the results interpreted appropriately.

Figure 3:

1. I find the order of this figure a little confusing. Starts with stem cell properties, jumps to chemoresistance, then back to stemness, the apoptosis and so on. I would suggest reordering.

2. Why has the apoptosis assay in D only been done using the K520? This should be done also with the K180 to show that the results stand true in different cell lines.
3. The authors use shRNA to deplete expression of ITGA7 in both the K180 and K520 cells (which does not look that impressive in the K520 western blot), and then only use 1 shRNA from the K520 to do the apoptosis assay in H. They then use 1 shRNA from each of the K180 and K520 in figure I, and are back to all in figure J. This approach makes me concerned that the data is not reproducible in all lines. Please provide data using both shRNA from both cell lines for the assays presented in H and i.
4. Using the shRNA cell lines the authors must do the assay the performed in E to look at tumour growth from small numbers of cells. If depleting ITGA7 results in regulation of stem cell genes then does it alter the ability of these cells to grow tumours from small numbers of tumour cells. This is very important the functional relevance of ITGA7.

Figure 4:

1. In A, ITGA7 is over expressed in two cell lines but then these are used selectively in the various assays presented in the figure. Again, all assays must be done with both lines.
2. The data presented in H is in my opinion irrelevant. It would be more informative if the authors were to look at tumour growth with limiting dilutions of cells as in figure 3e. In 4a they suggest that expression of ITGA7 results in upregulation of stem cell genes, therefore does it result in stem cell like tumour growth properties? This must be investigated in order to determine whether ITGA7 is master a regulator of the stem cell phenotype or just a marker.

Figure 5:

1. Again a seemingly random choice of models throughout this figure. What is the rationale for this and why are the assays not done with multiple models?
2. I don't find the EMT marker data presented in D and E that convincing. E-cadherin and beta-catenin do not regulated in all samples. I think these markers in particular should be supported with immunofluorescence staining as redistribution within the cell is more important than protein expression level.

Figure 6:

1. Again we are missing cell lines here. Why have these been left out?
2. Would be informative to see Src and phospho-Src included in the panel of western blots.
3. In my opinion Y15 is not a clinically relevant FAK inhibitor and should not be used to target FAK for these studies. Clinically relevant and specific FAK inhibitors are available commercially and should be used here. Both those from GSK and Verastem are available, although VS-4718 from Verastem is a dual FAK/Pyk2 inhibitor. The GSK inhibitor is FAK specific and the work should be done using this.
4. Much of the data presented here is already known for FAK.
5. If targeting FAK regulates cancer stem cell genes then what impact does it have on the ability of these cells to grow as tumours from low cell numbers? In ITGA7 cells, is FAK required for the stem cell like properties. At the very least this should be tested using a clinically relevant FAK inhibitor, or shRNA / CRISPR targeting FAK should be used to deplete FAK expression and the impact of this assessed with respect to Stem cell properties. This is important as FAK inhibitors are being developed as agents that target cancer stem cells.

A point-by-point response to the Reviewers' comments and suggestions

Reviewer #1:

The authors identify integrin Alpha 7 (ITGA7) as a putative CSC marker in esophageal squamous cell carcinoma. Previously, this group had identified CD 90 as a putative cancer stem cell marker and they now present evidence that a sub-fraction of CD 90 expressing cells also express ITGA7 and that these cells are enriched for CSC properties. Furthermore, they present evidence suggesting that ITGA7 regulates CSC properties via activation of focal adhesion kinase (FAK) and downstream signaling pathways including ERK, MAPK and AKT. In general, the experiments are well done and reported providing evidence for a functional role of ITGA7 in the stem cell phenotype of esophageal squamous cell carcinoma cells. However, a number of issues need to be addressed:

1. The authors initially identified ITGA7 as a putative CSC marker by virtue of its higher non GC methylation frequency. Based on this they speculate that ITGA7 might be activated by a non GC methylation (figure 1E). However, they only provide correlative evidence and there is no direct evidence for regulation of ITGA7 by non GC methylation in esophageal cancer stem cells. The authors either need to perform functional expression studies or acknowledge that their data are only correlative.

Our replay:

We agree with Reviewer's point. To determine whether the expression of ITGA7 could be restored by demethylation, a DNA methylation inhibitor (5-aza-2'-deoxycytidine) was used to treat sorted ITGA7⁺ and ITGA7⁻ cells. After treatment, expression of ITGA7 was dramatically increased (see revised Fig. 1e). According to reviewer's suggestion, we acknowledged in the revised manuscript that " ITGA7 expression might be correlated with non-CG methylation " (The 1st paragraph at page 6).

2. The most definitive assay for demonstration of tumor initiating capacity is the ability to form tumors in immunosuppressed mice. The authors do provide such data in figure 3B demonstrating the increased tumor initiating capacity of ITGA7⁺ compared to ITGA7⁻ cells. The authors need to calculate tumor initiating frequency utilizing extreme limiting dilution analysis. In addition, rather than just examining ITGA7⁺ versus ITGA7⁻ cells the authors should directly compare the tumor initiating frequency of ITGA7⁺ cells to CD 90⁺ ITGA7⁻ cells rather than all ITGA7⁻ cells. This is important since the authors claim that ITGA7 is a

more specific marker of CSC's than CD 90. The manuscript could be further strengthened if similar in vivo experiments were performed on ITGA7 knockdown cells or ITGA7 overexpressing cells. Although Figure 4H, which shows a higher growth rate and larger median tumor volume in tumors overexpressing ITGA7 is interesting, these characteristics are more reflective of growth rate of transient amplifying cells than of actual tumor initiating capacity. As was done in Figure 3.E the authors need to perform tumor initiating frequency studies.

Our replay:

In accordance with reviewer's suggestion, we performed similar *in vivo* experiments on ITGA7 overexpressing cells and ITGA7 knockdown cells, and calculate tumor initiating frequency utilizing limiting dilution analysis (Revised Fig. 3e, Fig. 4h&4i, and Supplementary Fig 3b&3c.). We agree with reviewer's suggestion, it is better to directly compare the tumor initiating frequency of ITGA7⁺ cells to CD90⁺/ITGA7⁻ cells. Based on our experience, it takes around 6 months to do cell sorting and *in vivo* experiment for CD90⁺/ITGA7⁺ and CD90⁺/ITGA7⁻ cells. So, we cannot complete this experiment within 3 months. According to our current and previous data, the tumor initiating frequency of ITGA7⁺ KYSE520 cells (1/46,830) is higher than that of CD90⁺ KYSE520 cells (1/57,076, Tang et al., Cancer Res, 2013), suggesting that CD90⁺/ITGA7⁺ KYSE520 cells possess stronger tumor initiating capacity than CD90⁺/ITGA7⁻ KYSE520 cells.

3. The authors demonstrate that ITGA7 knockdown reduces FAK activation. Other investigators have demonstrated that Beta-1 integrins are crucial regulators of FAK phosphorylation. The other tumor types alpha-6 beta-1 which is also a laminin receptor activates focal adhesion kinase through interactions with beta-1 integrin. Have the authors performed any experiments to elucidate whether hetero-dimerization of ITGA7 to beta-1 integrin is necessary to activate focal adhesion kinase?

Our replay:

This is a very important question. Actually, we performed qRT-PCR and IHC to detect $\beta 1$ and $\alpha 6$ integrins as they are reported as important CSC markers in some cancers. The results found that they are pervasively expressed in both ESCC and non-tumor tissues. Therefore, we did not study $\beta 1$ and $\alpha 6$ integrins in the present work.

4. The authors provide convincing evidence that ITGA7 activates focal adhesion kinase and

that a specific FAK auto phosphorylation inhibitor Y15 abolishes ITGA7 induced tumorigenesis and metastasis. Although they present western blotting showing that FAK activation results in downstream activation of MAPK and ERK, they provide no functional evidence that this downstream pathway mediates ITGA7 biological activities. The authors need to perform experiments utilizing MAPK and/or ERK inhibitors to show the importance of this pathway by demonstrating that these inhibitors abrogate the effect of ITGA7 overexpression.

Our replay:

In accordance with reviewer's suggestion, we treated cells with MAPK inhibitor U0126 and then did functional studies to conform that MAPK/ERK is the downstream player of ITGA7/FAK. The results showed that U0126 treatment significantly suppressed foci formation, cell migration and invasion abilities in ITGA7-transfected cells. This new data has been included in the revised manuscript (The 1st paragraph at page 13, Supplementary Fig. 5).

5. The authors compare ITGA7 expression in clinical ESCC specimens of patients who either have received chemotherapy before esophagectomy or had esophagectomy in the absence of chemotherapy and demonstrate that patients with preoperative chemotherapy had significantly higher percentage of ITGA7+ cells. While these results are interesting they can potentially be biased by patient selection as to which patients were eligible to receive preoperative chemotherapy. Is it possible that patients were selected to receive preoperative chemotherapy based on their more aggressive disease and that this could account for the increase in ITGA7 expressing cells? The authors should comment on potential biases introduced by patient selection.

Our replay:

In this experiment, 22 clinical specimens received preoperative chemotherapy were randomly collected from ESCC patients at the Queen Mary Hospital (Hong Kong), where preoperative chemotherapy was a standard practice. Twenty-eight clinical specimens without preoperative chemotherapy were randomly collected from ESCC patients at Sun Yat-Sen University Cancer Center (Guangzhou, China), where preoperative chemotherapy was not a standard practice. As Hong Kong is geographically close to Guangzhou, these two groups of samples were compared in this study. Therefore, we believe that the higher percentage of ITGA7⁺ cells in patients with preoperative chemotherapy would not be introduced by patient selection.

Reviewer #2:

The manuscript "Integrin $\alpha 7$ is a functional cancer stem cell marker in esophageal squamous cell carcinoma" by Ming et al., describes a role for Integrin $\alpha 7$ in cancer progression. The authors propose that a subpopulation of ITGA7⁺ cells in ESCC correlates with aggressive tumor behavior, poor outcome and cancer stemness and thus suggest ITGA7 is a CSC marker. Although it has been shown in a broad range of cancers that cell surface integrins including ITGA7 activate signaling pathways which lead to cancer stemness. In particular, ITGA7 has been linked to stemness of glioblastoma and mesenchymal cells. Also, the integrin signaling pathway they describe has already been linked to cancer aggressiveness and stemness in previous published work. To prove that ITGA7 expressing cells have cancer stem cell properties, the authors use ESCC cell lines and various stem cell assays; however the results do appear somewhat preliminary and not particularly unexpected based on previous reports.

Specific comments:

1) The major weakness of the paper is the novelty, as ITGA7 has been associated with stem cells in other tissues (mesenchymal) and cancers (glioblastoma) and this integrin signaling pathway is established and has been shown to result in a more metastatic and aggressive cancer. However, this is the first report linking ITGA7 as a biomarker for ESCC stemness. This is based the evaluation of cultured cell lines. To make their findings more relevant to the clinical course of ESCC the authors should examine patient derived xenograft (PDX) model for tumor initiation and the link to ITGA7. At least this approach would improve the relevance of their findings in terms of tumor cell stemness and cancer progression.

Our replay:

We understand reviewer's concern about the novelty of the work. The major contribution of this work is to demonstrate that ITGA7 is not only a CSC marker but also a key factor for CSC maintenance. Actually, this is one of the key issues in CSC research. So far as we know, none of the currently known CSC markers of ESCC has such property, i.e. to play key roles in CSC maintenance.

We agree with reviewer's suggestion that PDX model would improve the relevance of our findings in terms of ITGA7's role in tumor initiation. However, we cannot perform this experiments for two reasons: (1) the time limitation (we cannot complete it within 3 months); and (2) ITGA7⁺ cells were extremely low (<1%) in primary ESCC and PDX. In the present

study, we have mentioned that we could not to sort ITGA7⁺ cells from the primary ESCC tumors because the cell number limitation.

2) The author's aim is to demonstrate that ITGA7⁺ ESCC cells possess CSC properties. Using cell lines, they perform stem cell assays both in vitro and in vivo to determine ITGA7⁺ stem properties. Using FACS sorting they show ESCC cell lines express ITGA7 in a small sub population of cells. These cells also co-express cancer stem marker CD90 (previously published by the group) as shown in multiple panels in Figure 1. In order to prove ITGA7 expression is involved in cancer stemness they performed a knockdown of ITGA7 using shRNA (shITGA7). However the comparison should not be between control and shITGA7 of the entire cell line population since the majority of the naïve cells are already ITGA7⁻ (Figure 3h,I and Figure 5b). The experiment should be done by analyzing the CD90⁺ subpopulation with and without ITGA7 knockdown to show a differential effect on CSC properties. Similar sorting for CD90 positive cells should be done prior to western blot analysis using ITGA7 knockdown(Figure 3g,j and Figure 5E). The knockdown was only partial with a 50% decrease in expression of an already small population (22-25%). So it would seem analysis of ITGA7 knockdown in the subpopulation of CD90 expressing cells, which the authors claim co-express ITGA7, would be a better approach.

Our replay:

Thanks for reviewer's comment. However, it is hard to perform this experiment because of technical limitation. CSC culture is still a big problem in CSC research. For example, if we culture FACS-sorted CD90⁺ cells (90% in purity) from KYSE180 (original CD90⁺ proportion is 20%) for 1-2 week, the CD90⁺ cells would be reconstituted the original proportion (e.g. from 90% drop to 20%). In the present study, we used stable ITGA7 knockdown strategy and it takes 3-4 weeks. Therefore, we cannot perform the reviewer suggested experiment.

3) Figure 1B the authors do not describe where the data for the KM plot was obtained from. Is this from the 262 primary ESCC cases (Table 1)? If so do they use the same cut off value (0.6%) for high and low expression to represent ITGA7⁻ and ITGA7⁺ in the survival plot? Please clarify in results, figure legend or methods section.

Our replay:

Yes, the data for the KM plot was obtained from the 262 primary ESCC cases, and the same cutoff value was used. We described it in results section of the revised manuscript (The 1st paragraph at page 5).

4) In figure 1E, the authors show non cg-methylated sites in the ITGA7 gene loci of ITGA7+ cells, implying that ITGA7 expression might be activated by non-CG-methylation. Although interesting, other than this one piece of data (Figure 1E) they do not further elaborate or show the relevance of non-CG-methylation in ESCC tumor aggressiveness, stemness, overall patient outcome or ITGA7 expression. To strengthen this claim, further analysis is required. For example, studies could be performed using DNA methylase inhibitors to show an effect on ITGA7 expression. Alternatively, they should show a correlation between poor overall survival and non-methylated ITGA7 in ESCC patient tumors. Otherwise the data provided are not particularly relevant to the rest of the study.

Our replay:

In accordance with reviewer's suggestion, we used a DNA methylase inhibitor (5-aza-2'-deoxycytidine) to treat sorted ITGA7⁺ and ITGA7⁻ cells to determine whether ITGA7 expression could be increased. The results showed that the expression of ITGA7 could be dramatically increased after 5-aza treatment. This new data has been included in the revised manuscript (The 1st paragraph at page 6, revised Fig. 1e).

Reviewer #3:

In the manuscript by Ming et al, the authors identify integrin $\alpha 7$ (ITGA7) as a novel marker of a cancer cell population with stem-like properties. They further go on to show that this population of cells is more resistant to chemotherapy (5FU and cisplatin), and that they exhibit increased migration and invasion. Finally, they propose that the mechanism through which ITGA7 regulates these cellular properties is through activation of FAK/MEK/ERK signalling.

I think that aspects of this study are interesting. However, I feel that the authors have not gone far enough in addressing the pertinent questions arising from the work they present. When finished reading the manuscript I am left wondering whether ITGA7 is merely a marker of cancer stem cells (CSC) or is it functionally relevant in the regulation of the CSC phenotype. The link between integrins, FAK etc and chemoresistance, increased migration and invasion is in no way surprising or indeed novel. Therefore, I believe the authors need to show more definitively that ITGA7 not only marks CSCs but also plays an important role in regulating their phenotype and stem-like properties, and that FAK plays an important role downstream. The properties of these cells in terms of chemoresistance, invasion etc should

come after this to describe the importance of these cells in terms of therapy.

Our replay:

In the present study, we tried to demonstrate that ITGA7 is not only a ESCC CSC marker, but also plays important roles in CSC maintenance. First, we demonstrated that ITGA7⁺ cells could be sorted from ESCC cell lines, which possessed CSC properties, suggesting that ITGA7 could be used as CSC marker. Second, we demonstrated when ITGA7 was knocked down in ESCC cells, it decreased the stemness of cancer cells, suggesting that ITGA7 played roles in CSC maintenance.

My specific concerns are as follows:

Figure 1:

1. The immuno-staining in the top left panel of figure 1A does not look to have worked very well (large white area). Either this is a poor choice of image or the staining quality is poor and should be repeated.

Our replay:

After adjustment of the brightness, the staining looks better. In the top left picture of Fig. 1a, most of the cells are differentiated suprabasal cells which undergo multiple stages of differentiation from basement membrane, losing their nucleus, and then resulted in large white area.

2. In the text it is stated that based on frequency of ITGA7+ cells in the staining the samples have been divided up into high and low frequency group. This is based on what seems like a somewhat arbitrary cutoff of 0.6%. Is there a valid reason for this cutoff? If so, this should be stated in the text. Otherwise, what led to this choice of cutoff?

Our replay:

Here we tried to equally divide the clinical specimens of this TMA into two groups. Based on the frequency of ITGA7⁺ cells, we found 0.6% is a good cutoff value (>0.6%, n=137, 52.3%; ≤0.6%, n=125, 47.7%; see supplementary Table 1). We described it in results section of the revised manuscript (The 1st paragraph at page 5).

3. In figure 1C I would like to see a plot for each of the cell lines (although this could supplementary) and statistics comparing the data for the two normal cell lines to the 10 cancer lines. Also, immunofluorescence data presented in d does not really support the

frequencies presented in c, and I am not sure how useful this data really is?

Our replay:

In accordance with reviewer's suggestion, we displayed representative images of ITGA7% detected by FACS in all esophageal cell lines and the statistics analysis (see revised Fig. 1c and Supplementary Fig. 1a). In the Fig. 1d (moving to Supplementary Fig. 1b), we tried to confirm the ITGA7 expression level by immunofluorescence. The result showed that it is hard to detect ITGA7⁺ cells in NE1 and EC109, while around 10-20% ITGA7⁺ cells were detected in KYSE180 and KYSE520, which was consistent with FACS result.

Figure 2:

1. In figure 2a the gating used is different to that used in the other cell lines and there is no valid reason provided for this. Also, I want to see a scatter plot with gating for every cell line plus controls. When looking at the NE1 scatterplot it appears that there are two populations, a CD90- and a CD90+, and these would be apparent if the same gating had been used. It looks like there is a CD90+ population in the NE1 and that this population may acquire ITGA7 expression in the cancer cells. This would change the interpretation of the authors results presented in B. The authors should provide all of the data on which these conclusions are based, including controls, and if the gating is significantly different justify why. Otherwise the gating should be changed and the results interpreted appropriately.

Our replay:

In accordance with reviewer's suggestion, we provided dual color flow cytometry dot plots of ITGA7 and CD90, and related isotype control in all esophageal cell lines (Supplementary Fig. 2). The gating used is based on related isotype control.

Figure 3:

1. I find the order of this figure a little confusing. Starts with stem cell properties, jumps to chemoresistance, then back to stemness, the apoptosis and so on. I would suggest reordering.

Our replay:

Studies have now demonstrated that CSCs exhibit many classical properties of both normal stem cells and cancer cells, including the following: (i) a high self-renewal capacity; (ii) an enhanced ability to differentiate; (iii) an increased capacity for self-protection against drugs, toxins and radiation; and (iv) an increased capacity to initiate and sustain tumor growth. In Fig. 3a-e, we tried to confirm ITGA7⁺ ESCC cells possess these CSC properties, while in

Fig.3f-j, we found knockdown of ITGA7 could decrease cancer stemness properties in ESCC cells.

2. Why has the apoptosis assay in D only been done using the K520? This should be done also with the K180 to show that the results stand true in different cell lines.

Our replay:

We agree with reviewer's suggestion and performed apoptosis assay with ITGA7⁺ K180 cells and ITGA7⁻ K180 cells. This part of work has been incorporated into the revised Figure 3d.

3. The authors use shRNA to deplete expression of ITGA7 in both the K180 and K520 cells (which does not look that impressive in the K520 western blot), and then only use 1 shRNA from the K520 to do the apoptosis assay in H. They then use 1 shRNA from each of the K180 and K520 in figure I, and are back to all in figure J. This approach makes me concerned that the data is not reproducible in all lines. Please provide data using both shRNA from both cell lines for the assays presented in H and i.

Our replay:

In accordance with reviewer's suggestion, we performed apoptosis assay and spheroid formation assay in both cell lines and both shRNAs (see revised Fig. 3h-3i and Supplementary Fig. 3a). For the expression of ITGA7 in the K520 western blot, we replaced it with a less exposed blot of ITGA7 which looks better (see revised Fig. 3g).

4. Using the shRNA cell lines the authors must do the assay the performed in E to look at tumour growth from small numbers of cells. If depleting ITGA7 results in regulation of stem cell genes then does it alter the ability of these cells to grow tumours from small numbers of tumour cells. This is very important the functional relevance of ITGA7.

Our replay:

We totally agree with reviewer's comment, thus, similar in vivo experiment just as isolated ITGA7⁺ cells and ITGA7⁻ cells from KSYE180 and KYSE520 cells was carried out with ITGA7-suppressed cells. In order to minimize the number of mice to be used, two ITGA7-specific shRNAs treated cells were equally mixed together as shITGA7 group. Compared to control group, knockdown of ITGA7 effectively suppressed tumor initiation and progression. This new data has been included in the revised manuscript (The 1st paragraph at page 9, Supplementary Fig. 3b-3c).

Figure 4:

1. In A, ITGA7 is over expressed in two cell lines but then these are used selectively in the various assays presented in the figure. Again, all assays must be done with both lines.

Our replay:

In accordance with reviewer's suggestion, we performed apoptosis assay in EC109 (see revised Fig. 4c).

2. The data presented in H is in my opinion irrelevant. It would be more informative if the authors were to look at tumour growth with limiting dilutions of cells as in figure 3e. In 4a they suggest that expression of ITGA7 results in upregulation of stem cell genes, therefore does it result in stem cell like tumour growth properties? This must be investigated in order to determine whether ITGA7 is master a regulator of the stem cell phenotype or just a marker.

Our replay:

We agree with reviewer's comment, therefore, we performed *in vivo* experiment to look at tumor initiation potential with limiting dilutions of ITGA7-transfected cells and negative control cells. The result showed that ITGA7-overexpressed EC109 and KYSE30 cells displayed significantly stronger tumor initiation ability than their control cells. This new data has been included in the revised manuscript (The 2nd paragraph at page 10, revised Fig. 4h-4i).

Figure 5:

1. Again a seemingly random choice of models throughout this figure. What is the rationale for this and why are the assays not done with multiple models?

Our replay:

Thanks for reviewer's reminder. Actually, we performed not just these experiments showed in Fig.5a-5b to demonstrate the effect of ITGA7 on cell mobility. First, we performed migration and invasion on isolated ITGA7⁺ cells and ITGA7⁻ cells from both KSYE180 and KYSE520 cells (Fig. 5a and Supplementary Fig. 4a) to show that ITGA7⁺ cells have higher ability to promote cell migration and invasion. Then, we performed wound healing assay in both ITGA7-overexpressed cell lines and both ITGA7-suppressed cell lines (Supplementary Fig. 4b) to display that ITGA7 could dramatically enhance tumor cell mobility. For further confirmation, we chose one cell line for migration and invasion assay (Fig. 5b and

Supplementary Fig. 4c). This new data has been included in the revised manuscript (The 1st paragraph at page 11)

2. I don't find the EMT marker data presented in D and E that convincing. E-cadherin and beta-catenin do not regulated in all samples. I think these markers in particular should be supported with immunofluorescence staining as redistribution within the cell is more important than protein expression level.

Our replay:

We agree with reviewer's suggestion. From the immunofluorescence staining result (see revised Fig. 5d), expression of E-cadherin was decreased in *ITGA7*-transfected cells compared with control cells. Although the expression level of β -catenin remained unchanged, its nuclear accumulation was observed in *ITGA7*-transfected cells. EMT markers were also detected in *ITGA7*-transfected cells by western blotting (see revised Fig. 5e). This new data has been described in the revised manuscript (The 2nd paragraph at page 11).

Figure 6:

1. Again we are missing cell lines here. Why have these been left out?

Our replay:

Please see revised Fig. 6b. We detected the phosphorylation of FAK/MAPK/ERK signaling pathway in EC109 cell line, which is consistent with KYSE30.

2. Would be informative to see Src and phospho-Src included in the panel of western blots.

Our replay:

According to reviewer's suggestion, we detected total Src and phospho-Src by western blot (Revised Fig. 6a-6c).

3. In my opinion Y15 is not a clinically relevant FAK inhibitor and should not be used to target FAK for these studies. Clinically relevant and specific FAK inhibitors are available commercially and should be used here. Both those from GSK and Verastem are available, although VS-4718 from Verastem is a dual FAK/Pyk2 inhibitor. The GSK inhibitor is FAK specific and the work should be done using this.

Our replay:

Although Y15 is not a clinically relevant FAK inhibitor, Y15 can specifically block Y397-FAK autophosphorylation which is the main phosphorylation site activating FAK signaling in the cells by *in vitro* study (Golubovskaya et al., J Med Chem, 2008). To further confirm FAK/MAPK/ ERK signaling mediates ITGA7 biological activities, we treated cells with MAPK inhibitor U0126 and then did functional studies to conform that MAPK/ERK is the downstream players of ITGA7/FAK. This new data has been described in the revised manuscript (The 1st paragraph at page 13, Supplementary Fig. 5).

4. Much of the data presented here is already known for FAK.

Our replay:

We performed these experiments to demonstrated that ITGA7 regulated CSC properties in ESCC through FAK signaling pathway.

5. If targeting FAK regulates cancer stem cell genes then what impact does it have on the ability of these cells to grow as tumours from low cell numbers? In ITGA7 cells, is FAK required for the stem cell like properties. At the very least this should be tested using a clinically relevant FAK inhibitor, or shRNA / CRISPR targeting FAK should be used to deplete FAK expression and the impact of this assessed with respect to Stem cell properties. This is important as FAK inhibitors are being developed as agents that target cancer stem cells.

Our replay:

In the present study, we found that ITGA7 plays an important role in CSC maintenance via FAK signaling pathway. It is well known that FAK signaling is critical in driving a number of biological processes including cell proliferation, differentiation and cell motility. Here we find that FAK is a versatile and important downstream player for ITGA7 mediated biological activities. Application of FAK inhibitor Y15 confirms that the regulatory effect of ITGA7 on CSC maintenance is FAK dependent.

REVIEWERS' COMMENTS:

Reviewer #1 (Remarks to the Author):

The authors have addressed my concerns and the revised manuscript is considerably improved.

Reviewer #2 (Remarks to the Author):

The authors determined that hypothetically my two major concerns would either take too much time or be too difficult to address. I respectfully disagree with their assessment. The novelty of this work and its clinical relevance remains a concern. My suggestion is that the authors make an attempt to address my original points. Use of PDX for clinical relevance and the sorting of the CD90 positive cells for ITGA7 knockdown. Without this data I believe these studies remain considerably less compelling. I have suggested some reasonable approaches the authors could use to complete these studies.

Reviewer #3 (Remarks to the Author):

The authors have addressed the majority of my concerns. The experiments appear to be well done. The only concern I have remaining is regarding figure 5. The regulation of E-cadherin appears cell line specific (perhaps this needs to be commented on in the text) and the nuclear accumulation of beta-catenin is not particularly convincing in the immunofluorescence pictures. Apart from that I have no further comments.

A point-by-point response to the Reviewers' comments and suggestions

Reviewer #1:

The authors have addressed my concerns and the revised manuscript is considerably improved.

Reviewer #2:

The authors determined that hypothetically my two major concerns would either take too much time or be too difficult to address. I respectively disagree with their assessment. The novelty of this work and its clinical relevance remains a concern. My suggestion is that the authors make an attempt to address my original points. Use of PDX for clinical relevance and the sorting of the CD90 positive cells for ITGA7 knockdown. Without this data I believe these studies remain considerably less compelling. I have suggested some reasonable approaches the authors could use to complete these studies.

Reviewer #3:

The authors have addressed the majority of my concerns. The experiments appear to be well done. The only concern I have remaining is regarding figure 5. The regulation of E-cadherin appears cell line specific (perhaps this needs to be commented on in the text) and the nuclear accumulation of beta-catenin is not particularly convincing in the immunofluorescence pictures. Apart from that I have no further comments.

Our replay:

We agree with Reviewer's suggestion. So we described that the regulation of E-cadherin may be cell line specific in the revised manuscript (The 1st paragraph at page 12). For the nuclear accumulation of beta-catenin in the K30-ITGA7 cells, we replaced it with a more convincing immunofluorescence picture (see revised Fig. 5d).